# An examination of force maps targeted at orientation interactions in moving groups

**Rajnesh K. Mudaliar**[1,2]*, **Timothy M. Schaerf**[2]

**1** School of Mathematical and Computing Science, Fiji National University, Ba, Fiji, **2** School of Science and Technology, University of New England, Armidale, NSW, Australia

* Rajnesh.Mudaliar@fnu.ac.fj

## Abstract

Force mapping is an established method for inferring the underlying interaction rules thought to govern collective motion from trajectory data. Here we examine the ability of force maps to reconstruct interactions that govern individual's tendency to orient, or align, their heading within a moving group, one of the primary factors thought to drive collective motion, using data from three established general collective motion models. Specifically, our force maps extract how individuals adjust their direction of motion on average as a function of the distance to neighbours and relative alignment in heading with these neighbours, or in more detail as a function of the relative coordinates and relative headings of neighbours. We also examine the association between plots of local alignment and underlying alignment rules. We find that the simpler force maps that examined changes in heading as a function of neighbour distances and differences in heading can qualitatively reconstruct the form of orientation interactions, but also overestimate the spatial range over which these interactions apply. More complex force maps that examine heading changes as a function of the relative coordinates of neighbours (in two spatial dimensions), can also reveal underlying orientation interactions in some cases, but are relatively harder to interpret. Responses to neighbours in both the simpler and more complex force maps are affected by group-level patterns of motion. We also find a correlation between the sizes of regions of high alignment in local alignment plots and the size of the region over which alignment rules apply when only an alignment interaction rule is in action. However, when data derived from more complex models is analysed, the shapes of regions of high alignment are clearly influenced by emergent patterns of motion, and these regions of high alignment can appear even when there is no explicit direct mechanism that governs alignment.

## 1 Introduction

Animal groups often engage in collective motion. Both theoretical models and empirical studies suggest that the patterns of collective motion arise due to local "rules of interaction" that govern how individuals adjust their velocity in response to the relative positions and behaviours of neighbours. Theoretical models often prescribe rules of interaction derived from some combination of three broad principles: individuals will adjust their velocity to (1) avoid

**Data Availability Statement:** All computer codes required to generate the data for this study (via simulation), and then analyse this data, have been provided with the submission.

**Funding:** This work was supported in part by funding from the Australian Research Council's Discovery Project scheme, under project code DP190100660. The funders had no role in study design, data collection and analysis, decision to publish, or preparation of the manuscript.

**Competing interests:** The authors have declared that no competing interests exist.

collisions with near neighbours (a repulsion/collision avoidance rule), (2) match their velocity with neighbours at some intermediate distance (an orientation/alignment rule, potentially including speed matching), and (3) adjust their velocity to move toward far neighbours (an attraction/cohesion/group joining rule).

Key to furthering understanding of collective motion has been the development of techniques for inferring associated rules of interaction directly from real world individual trajectory data, in a form that is directly comparable to hypothesised interaction rules applied in collective motion models. These methods include force-matching [1, 2], equation-based methods [3–9], force-mapping [2, 8, 10, 11] and identification of interactions via machine learning [12]. Force matching had its origin in molecular dynamics, [13–15], and is a form of regression analysis where a set of hypothesised forces is fitted to observed forces from experimental data by minimizing mean square differences [1]. Equation-based methods [3, 4, 6, 7, 9] have been developed with the goal of developing agent-based models where individual behavioural rules are informed directly by experimental data (the resulting models are sometimes referred to as data-driven models). The broad methodology involves proposing a set of equations and stochastic models for elements of movement that include the speed, turning behaviour and wall avoidance behaviour of individuals in isolation, along with repulsion, orientation and attraction interactions with neighbours. The parameters within the equations and models are determined via statistical fitting, and then characteristics of simulations of the resulting data-driven models are compared with experimental data for validation. Elements of both force-matching and the equation based methods bear some similarity to an approach applied by Lukeman et al. [16], with the main difference being how best fit was determined. In the case of Lukeman et al. [16], comparisons were made of model predicted and observed distributions of the relative positions and headings of neighbours, whereas best fit was determined via comparison of observed and predicted forces in force-matching and underlying measures of locomotion in the equation-based approach. Force-mapping methods [2, 8, 10, 11] examine the responses of individuals to their neighbours in terms of changes in velocity, without making an explicit hypothesis about the forms of these responses beyond the relevant independent variables. In the simplest form of force-mapping, the response is broken down into the means of the components of a change in velocity as a function of the independent variables, which are often the relative coordinates, and sometimes the components of velocity of the individuals and/or their neighbours. The machine learning approach is similar to that of force-mapping, in that it infers how individuals adjust their velocity in response to relevant independent variables, however the response is determined via training, rather than calculation of simple means. The machine learning method also includes a separate module to estimate the weights associated with interactions with multiple partners (the force-mapping method does not do this, and just aggregates all the data together).

The initial study that applied force-matching in the context of studies of collective motion used simulated, rather than experimentally derived, data. Eriksson et al. [1] simulated data of flocks where individual acceleration was controlled by a sum of forces including repulsion, attraction, velocity matching, friction (the tendency for individuals to decelerate in the absence of neighbours) and noise. Velocity matching and attraction forces could operate over metric (distance-based) or topological (distance-based neighbour rank) scales in the model. Hypothesised forces for the force-matching were structured in the same form as the model prescribed rules of individual acceleration, taking into account orientation and attraction interactions over metric scales only, over topological scales only, or both. The most reliable approach for identifying the weights for each term in the sum of hypothesised social forces was when both geometric and topological forces were fitted simultaneously using multiple regression.

Katz et al. [2], applied a variant of the force-matching method for the analysis of groups of golden shiners (*Notemigonus crysoleucas*). Rather than proposing a detailed model for how

individuals adjusted their velocity, as had been explored in Eriksson et al. [1], predicted social forces in the model were reduced to a single force tangential to an individual's direction of motion (termed a speeding force), and a single force perpendicular to the current direction of motion (termed a turning force). The unknowns to be fitted were the coefficients of the unit vectors in the directions of the speed and turning forces. These coefficients were treated as functions of the relative $(x, y)$ coordinates of neighbours, and discretised over a square grid, comprised of separate square bin regions and centred around a focal individual. The values of the coefficients were determined via a least squares process, as standard for force-matching. The resulting functions described how an individual adjusted its velocity as a function of the relative coordinates of neighbours, without explicitly breaking down contributions to these adjustments from potential underlying repulsion, orientation and attraction based behavioural rules. In addition, Katz et al. [2], and in a concurrent study, Herbert-Read et al. [10], introduced an alternative method for inferring interactions based around examining the mean changes in the components of velocity of individuals as a function of relevant independent variables (relative $(x, y)$−coordinates of neighbours, the speed of a focal individual, the speed of a neighbour, or the direction to a focal individual in some combination in Katz et al. [2] and Herbert-Read et al. [10]). Similar to the simplified force matching applied by Katz et al. [2], this analysis did not assume a set of equations that drove changes in direction due to underlying behavioural rules. The resulting functions fitted using this approach were referred to as "force maps" by Katz et al. [2], a naming convention that has persisted through some other studies that apply or examine the same analysis (for example, Zienkiewicz et al. [8], Escobedo et al. [9]). Herbert-Read et al. [10] did not name the function-fitting process, nor was the process named in subsequent works derivative of Herbert-Read et al. [10] (see for example, [11, 17, 18]). In broad terms, the characteristic of a force-mapping approach for estimating interaction rules is that these methods examine the response of individuals, in terms of changes in velocity, as a function of a number of predetermined independent variables, without an explicit hypothesis for the drivers of the individual response (other than the choice of independent variables). Data is usually aggregated across time and group configurations in producing force maps, and the approach has usually been to measure responses in terms of averages of the changes in the components of individuals' velocity, taking into account two or three independent variables. However, the machine learning approach applied by Heras et al. [12] can be viewed as a more sophisticated approach to generating force maps. This machine learning approach takes into account many independent variables, and applies a separate module to estimate the weights associated with interactions with multiple partners, something that has been ignored via the simpler approach. Force-mapping has also been used as an initial method of analysis to inform an equation-based data driven model for collective motion. Zienkiewicz et al. [8] used force maps to examine changes in the components of velocity, and angular acceleration, as a function of the relative coordinates and differences in relative directions of motion of neighbours in groups of zebrafish. The results of the analysis were used to inform a detailed model of the collective movements of zebrafish, that built on a previous model by Zienkiewicz et al. [4] for solitary fish taking into account interactions with boundaries, and that adopted a similar equation-based approach to that of Gautrais et al. [3], with model parameters fitted from the data (using maximum-likelihood estimates).

The equation-based approach for inferring interactions developed in parallel with the work by Lukeman et al. [16], the application of force-matching and the development of force maps/ force mapping, and has been largely focussed on studies of fish. In initial work, Gautrais et al. [5] analysed trajectories of individual nine barred flagtails (*Kuhlia mugil*) swimming in an arena to develop a model that characterised the spontaneous movement of a single fish, including wall avoidance. The swimming behaviour of the fish was observed as a combination of

spiral course, wall following, wall attraction and wall avoidance. Individual turning speeds estimated from the experimental observation were compared with output from an Ornstein-Uhlenbeck stochastic differential equation (referred to as a persistent turning walker (PTW) model). A valid PTW model for the fish was characterised by a constant swimming speed and an autocorrelation of the turning angle per unit time (heading) with an additional term added that bent the fish's trajectory away from the wall. Gautrais et al. [3] extended the previous work in [5] to include social rules of interaction, and applied the extended analysis to trajectory data from experiments with pairs and groups of five fish. The model developed via the analysis was a stochastic differential equation (SDE) that had a built-in balancing mechanism between individuals' positions, orientation, speed and a topological interaction neighbourhood, with near neighbours belonging to the first shell in a Voronoi neighbourhood. The parameters of the stochastic differential equation were obtained from the trajectory data using a standard non-linear least squares procedure. In order to validate the stochastic differential equation model a statistical comparision was performed using the mean and 95% confidence interval of group polarization (a measure of the degree of alignment among individuals in a group) and neighbour distances calculated from model simulated data and experimental data. Gautrais et al. [3] identified behavioural change (a transition from schooling to shoaling) among fish when the density of fish increased. Zienkiewicz et al. [4] applied similar analysis to that developed across [3, 5], and in initial work identified that the PTW model with constant speed as in [5] cannot be applied to suitably describe the behaviour of zebrafish *(Danio rerio)*. Zienkiewicz et al. [4] extended the PTW model in [5], to allow for the regulation in speed observed in zebrafish with the resulting model consisting of two SDEs, one for speed and one for turning speed. In all the three studies [3–5] the interactions that led to the development of an SDE model were assumed to take the form of a reasonably chosen function whose parameters were estimated from experimental data. Following a similar sequence of development to that of Gautrais et al. [3, 5], Zienkiewicz et al. [8] then extended the model for the movement behaviour of zebrafish to take into account social interactions, with the form of the equations used in the proposed model informed by an initial analysis of interactions via force maps, as discussed in the previous paragraph. Calovi et al. [7] extended the work by Gautrais et al. [3] to develop a stochastic model for the spontaneous burst-and-coast swimming behaviour of single Rummy-nose tetra *(Hemigrammus rhodostomus)* and pairs of Rummy-nose tetra. The SDE model consisted of sum of response functions of fish governing their heading changes as a function of the distance, orientation, and angular position relative to an obstacle or a neighbour. Each of these functions were directly estimated from experimental data using the method of least square regression analysis. In related work, Harpaz et al. [6] analysed the behaviour of individual zebrafish in groups and identified that fish alternate between an "active" mode, in which they are assumed to be sensitive to the swimming patterns of conspecifics, and a "passive" mode, where they are assumed to ignore them. These observations were integrated into a data-driven model.

As part of an examination of both force maps and equation-based/data driven model methods, Escobedo et al. [9] identified some of the potential limitations in using force maps to examine underlying behavioural rules. These limitations were listed as: (i) force-maps can handle a limited number of state or independent variables (in practice this has been two or three independent variables, often including the relative coordinates of neighbours in one- or two-dimensions [2, 8–11]); (ii) there is no explicit mechanism in the calculation of force maps to disentangle the underlying behavioural rules that contribute to an observed adjustment to velocity, such as orientation and attraction rules; (iii) the results of force mapping do not lead directly or easily to a mathematical model; (iv) it is difficult to distinguish between the effects of variables and parameters in terms of individual responses identified in force maps. The

above potential limitations were examined via reasonable qualitative arguments, but were not further interrogated quantitatively. One way to further investigate the accuracy of force maps is to examine the ability of such analysis to reflect model-based interaction rules from simulated data. Such an approach was applied by Eriksson et al. [1] to the original force matching method designed for collective motion analysis, and more recently by Heras et al. [12]. In lieu of experimental data at the time of their study, Eriksson et al. [1] trialled their force-matching approach for inferring rules of interaction using individual movement data simulated by a zonal model. In effect, the analysis that they performed acted as a form of validation for the method when reasonable hypotheses about the underlying rules of interaction could be made. Heras et al. [12] similarly validated their machine learning based method for inferring rules of interaction using data simulated by a zonal model similar to that of Aoki [19] and Couzin et al. [20].

In spite of now being one of the older methods used to examine rules of interaction, having been in use for just over ten years, there has only been one study that has attempted to expicitly validate force mapping and better understand the inaccuracies of the method. In [21] simulation data generated by two well-established individual-based models for collective motion, the zonal model described in [20] and the ordinary differential equation (ODE) model described in [22], was fed through standard force mapping calculations. The force maps that were constructed were best suited for identifying repulsion- and attraction-like behaviour governed by the relative positions of neighbouring group members. The maps generated by the calculations were then compared with those formed directly from prescribed model interaction rules for each model for simulation sets that varied across a number of measures. These measures included the number of individuals in each group, the presence and detail or absence of a blind region behind individuals, within model parameters that governed the strength and spatial extent of different rules of interaction, emergent patterns of motion, and the total number of time steps of each simulation. Broadly, the investigation suggested that force mapping is capable of revealing the qualitative form of interaction rules directly from trajectory data, including short range repulsion-like and longer range attraction-like behaviour governed by changes in speed and direction. The method also exhibited quantitative accuracy in revealing the spatial extent of the zone of repulsion from data generated by the zonal model, including the presence or absence of a blind zone, and the angular extent of this blind zone. The repulsion zone and effect in the zonal model is special in that it has priority over all other interactions, and is the only influence on how individuals adjust their velocity when neighbours occupy the corresponding region in space. However, several potential inaccuracies in force mapping were identified, in addition to those posited in [9], including sensitivity of the details of the force maps to group level patterns of motion, and variations in the quantitative details of the maps in concert with varying group size. In particular, the size of the region over which repulsion effects seemed to apply according to force maps derived from ODE model data shrank as group size increased. Unlike the zonal model, there is no priority given to repulsion/ collision avoidance in the ODE model, so the diminishing size of the repulsion zone may be due to individuals adjusting their velocity due to a sum of both repulsion- and attraction-like behaviour.

Although the tendency to align with near neighbours is thought to be a key driver of the emergent patterns of collective movement, direct analysis seeking these interactions via force maps has been more limited than analyses targeted at repulsion- and attraction-like behaviour, and there has been no validation of force mapping applied to alignment behaviour. The initial work by Herbert-Read et al. [10], included an examination of how individual eastern mosquitofish adjusted their direction of motion in response to the direction to neighbours, and differences in relative directions of motion with the same neighbours. The analysis in Herbert-Read

et al. [10] was subdivided to examine turning responses in what had been inferred to be the repulsion and attraction zones of the mosquitofish, and broadly seemed to suggest that this species did not have an underlying behavioural tendency for direct alignment with neighbours. Similar analysis was applied in a study of guppies (*Poecilia reticulata*) [17]. More recently, Zienkiewicz et al. [8] examined potential orientation interactions in shoals of zebrafish (*Danio rerio*) with force maps that detailed changes in direction of motion or angular acceleration as a function of several variables that included differences in relative directions of motion. The work in Zienkiewicz et al. [8] included the development of a data-driven model informed by the force mapping analysis, and subsequent examination of force maps of simulated, rather than real, behaviour. One of the revelations of this work was that force maps examining angular acceleration, rather than changes in direction, were better indicators of an underlying tendency for individuals to align with each other.

Here, we adapt the approach applied in [21] to examine the efficacy of force mapping in identifying explicit alignment behaviour from simulated trajectory data. We simulate collective motion using three well-established general models which differ in how, or whether or not, alignment behaviour is treated. These models are the self-propelled particle model described in [23], where alignment is the only interaction rule, and the zonal model and ODE model of [20] and [22], as used previously in [21]. Given that the rules for direction changes due to orientation interactions in the alignment-only model and the zonal model explicitly relate to changes in direction as a function of the distance to neighbours and differences in directions of motion between a focal individual and its neighbour, we modified the force mapping analysis applied in Mudaliar et al. [21] to target such an interaction via several approaches. In particular, we examined the ability of force maps that described changes in direction as a function of the relative $(x, y)$ coordinates of neighbours and differences in direction of motion, or as a function of the distance to the neighbour and differences in direction of motion to inform of the form and presence (or lack thereof) of orientation interactions.

In addition to force-maps examining changes in direction, another important tool for examining local alignment within groups in detail has been to construct plots that examine the mean directions of motion of neighbours as a function of the coordinates of these neighbours relative to a focal individual [16], along with a measure of the variance about these mean directions [11, 24–27]. As part of the work in this paper, we examine the association between such plots and underlying behavioural rules for orientation, again using data simulated by the zonal and ODE models as detailed in [21] and alignment-only models [23].

## 2 Analysis seeking orientation interactions and graphs of local alignment

The broad approach applied throughout this study was to generate trajectory data via simulation of collective motion from models with explicit known interaction rules, and then examine how well various force maps could reconstruct the underlying model simulation rules. This approach was applied previously in [21], but here we focus on orientation interactions, whereas the previous study focussed on repulsion/collision avoidance and attraction/cohesion interactions. We chose three very well established self-propelled particle models to generate our synthetic data sets—the model described by [23], where the only rule of interaction involves orienting directions of motion with neighbours, the more complex zonal model described by [20], where individuals are repelled by, orient with, or are attracted to neighbours over different ranges of distances, and the ordinary differential equation (ODE) model described by [22], where individuals are repelled by or attracted to neighbours, but there is no direct mechanism to force alignment.

In this larger section, we first give an overview of an existing approach for force mapping (with full details provided in the S1 File, Section S1 in S1 File), and then detail calculation of the vital variable for examining orientation interactions (the angular difference in directions of motion between individuals). In conjunction with the force maps seeking orientation interactions, we examine the association between orientation interactions and graphs of local alignment. We then provide an overview of the three models used, including the simulations performed, with full details of our implementation given in the supplementary information. Simulations were performed multiple times for different sets of parameter values. In some cases we segregated simulation output based on patterns of emergent motion. The reason for dividing data based on emergenet pattern of motion, where relevant, was that previous work in [21] suggested that emergent patterns of motion can affect the details of force maps, at least when considering repulsion- and attraction-based interaction rules. It is possible to construct an exact equation-based form for the interaction rules in response to a single neighbour for each model; we detail these for orientation interactions for the orientation-only and zonal models. The exact pairwise interactions can then be used as a point of comparison with the rules inferred via force mapping. Full details of the models, simulations performed and the exact forms for relevant pairwise interactions are provided in the S1 File, Section S2 in S1 File.

## 2.1 Force mapping

In one of its simplest forms, force mapping is a method for estimating the mean changes in the components of velocity of individuals as a function of the relative coordinates of neighbouring group members in a consistently oriented frame of reference. In prior studies, other standard independent variables that have been used include the speed of the "focal" individual [10, 11] and the speed of neighbours [2]. The necessary data for force mapping consists of discrete time series of the coordinates of each group member, indexed by $i$, usually in two dimensions $(x_i(t), y_i(t))$ at some discrete time $t$, with the data equally spaced in time with temporal separation $\Delta t$. Fundamental coordinate geometry and standard methods for the numerical estimation of derivatives are then applied to determine a number of locomotory and relative spatial measures. Locomotory measures often include individual speed, $s_i(t)$, direction of motion, $\theta_i(t)$, velocity, $\mathbf{V}_i(t) = u_i(t)\mathbf{i} + v_i(t)\mathbf{j}$ (where $\mathbf{i}$ and $\mathbf{j}$ are unit vectors in the $x$- and $y$-directions respectively), associated unit direction vector, $\hat{\mathbf{V}}_i(t) = \hat{u}_i(t)\mathbf{i} + \hat{v}_i(t)\mathbf{j}$, change in speed over time, $\frac{\Delta s_i}{\Delta t}(t)$, and change in direction of motion over time, $\frac{\Delta \theta_i}{\Delta t}(t)$. The latter two locomotory measures in combination are one way in which changes in the components of an individual's velocity can be described. Relative spatial measures include the distance from each individual, $i$, to its neighbour, $j$, denoted $d_{i,j}(t)$, the angle between the velocity vector of an individual $i$ and the straight line from that individual to its neighbour, $\vartheta_{i,j}(t)$, and the differences between the directions of motion of individuals $i$ and $j$, denoted here as $\varphi_{i,j}(t)$. The distance $d_{i,j}(t)$ and angle $\vartheta_{i,j}(t)$ can be used to determine the coordinates $(x, y)$ of individual $j$ relative to individual $i$, in a frame of reference where individual $i$'s velocity vector is aligned with the positive $x$-axis; this is one of the standard coordinate systems for the force maps examined here.

The construction of force maps involves binning a variable of interest, for example the change in speed over time of individual $i$ at a given time $t$, according to the relative coordinates of each neighbour $j$, with data aggregated across all focal individuals $i$, each of their neighbours $j$ and for all discrete times $t$. Once the binning process is complete, means of the values in each bin are determined; in the case of the change in speed over time of focal individuals $i$, this results in a function (stored as a lookup table/matrix) that describes how individuals adjust their speed on average in response to the relative $(x, y)$ positions of neighbours. Some

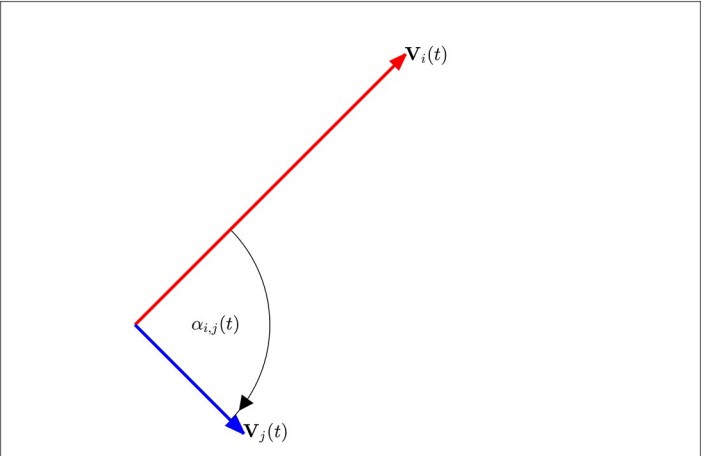

**Fig 1. Angular difference, $\varphi_{i,j}$, between the velocity vectors of individuals $i$ and $j$ at a given time $t$, formed when $\mathbf{V}_i(t)$ and $\mathbf{V}_j(t)$ are placed tail-to-tail (adapted from [26]).** $\varphi_{i,j}$ is positive when the rotation from $\mathbf{V}_i(t)$ to $\mathbf{V}_j(t)$ is anticlockwise, and negative when the rotation is clockwise (as pictured here).

smoothing is often applied to the force map, including by some form of linear interpolation applied to construct a graph of the function, through the use of overlapping bins during the averaging process, or via some other standard technique. Full details for constructing force maps of the mean change in speed and change in direction of motion of individuals as a function of the the relative $(x, y)$ coordinates of neighbours following the method used in [11, 21, 26] are provided in Section S1 of the S1 File. S1, S2 Figs in S1 File and Fig 1 illustrate many of the important quantities deduced from the coordinate data, including those noted above.

**2.1.1 Force mapping targetted at orientation rules.** We modified the force mapping method applied in [11, 21, 26] so that the distance to a neighbour ($d_{i,j}$) and the direction of motion of a neighbour relative to that of a focal individual ($\varphi_{i,j}$), detailed explicitly below and as illustrated in Fig 1, can be treated as independent variables. This particular combination of independent variables are those that often dictate orientation interactions in theoretical models, where focal individuals act to match velocity/align directions of motion with neighbours that occupy a particular range of distances, as is the case in the self-propelled particle models developed in [20, 23]. As noted in the introduction, prior studies have applied force mapping seeking orientation interactions [8, 10, 17], but not with this particular combination of independent variables.

The main expansion to the calculations described in the SI is the determination of the angular difference in directions of motion between a focal individual, $i$, and its neighbour, $j$, denoted $\varphi_{i,j}(t)$, as illustrated in Fig 1. This angular difference is given by:

$$\varphi_{i,j}(t) = \begin{cases} \gamma_{i,j}(t)\alpha_{i,j}(t) & \text{if } \gamma_{i,j}(t) \neq 0, \\ \\ \alpha_{i,j}(t) & \text{if } \gamma_{i,j}(t) = 0, \end{cases} \tag{2.1}$$

where the magnitude of the angular difference is

$$\alpha_{i,j}(t) = \cos^{-1}\left(\hat{u}_i(t)\hat{u}_j(t) + \hat{v}_i(t)\hat{v}_j(t)\right), \tag{2.2}$$

and the corresponding sense of rotation from the direction of motion of individual $i$ to $j$ is

given by

$$\gamma_{i,j}(t) = \text{sgn}\left(\hat{u}_i(t)\hat{v}_j(t) - \hat{u}_j(t)\hat{v}_i(t)\right), \tag{2.3}$$

where sgn is the sign function. Relative to the direction of motion of the focal individual $i$, the sense of rotation of individual $j$ is anticlockwise (clockwise) relative to the individual $i$ if Eq (2.3) is positive (negative).

Our orientation rule focussed force maps applied throughout this study examined how individuals adjusted their direction of motion, $\dfrac{\Delta\theta}{\Delta t}$, as a function of the relative $(x, y)$ coordinates of neighbours and angular difference in directions of motion $\varphi_{i,j}$ (that is, as a function of $(x, y, \varphi_{i,j})$), or as a function of just the distance to the neighbour, $d_{i,j}$ and the angular difference $\varphi_{i,j}$ (that is, as a function of $(d_{i,j}, \varphi_{i,j})$). Full details of the associated binning process are included in the SI, Section S1.4. We rendered our force maps using MATLAB's built-in *surf* function with interpolated shading.

**2.1.2 Local alignment graphs.** In concert with force maps targetted at orientation interactions, we examined the association between plots of local alignment, like those applied in [11, 24–26], which were derived from an initial approach in [16], and prescribed orientation rules. Explicitly, the local alignment plots illustrate the mean directions of motion of neighbours relative to the direction of motion of a focal individual (oriented with the positive $x$-axis) when the neighbours occupy given relative $(x, y)$ coordinates, and a measure associated with the deviation of the observed angular differences, termed the focus. We constructed the local alignment plots as an extension of the force mapping calculations outlined above, via the following additional calculations. We first binned angular differences, $\varphi_{i,j}$, into respective bins based on the relative $(x, y)$ coordinates of neighbours, with data aggregated across all times within each simulation, for all combinations of focal individual $i$ and neighbours $j$, and across all simulations within a distinct set. Then the mean, $\bar{\varphi}$, of a set of binned angles $\varphi_i$ (where the subscript denotes the $i$th element of a given bin in this case), was obtained via:

$$\bar{\varphi} = \text{atan2}\left(\left(\sum_{i=1}^{n}\sin\varphi_i\right), \left(\sum_{i=1}^{n}\cos\varphi_i\right)\right), \tag{2.4}$$

where $\text{atan2}(Y, X)$ is a common numerical function for correctly determining the angle and quadrant associated with displacements of $X$ in the $x$-direction and $Y$ in the $y$-direction, such that $-\pi < \bar{\varphi} \leq \pi$ (in radians). The measure $R$ defined below was used to quantify the focus of binned angles about the mean for each bin:

$$R = \frac{\sqrt{\left(\sum_{i=1}^{n}\cos\varphi_i\right)^2 + \left(\sum_{i=1}^{n}\sin\varphi_i\right)^2}}{n}, \tag{2.5}$$

where $R \in [0, 1]$, with values of $R$ closer to 1 indicating a lower scatter of angles about the mean (greater focus) [11, 28]. Values of $R$ are in the same range and are similar to polarisation, a measure used to quantify instantaneous alignment between group members in many collective motion studies. However, the two measures differ as the polarisation describes the instantaneous alignment of a group, whereas the binned angles are accumulated across a variety of time steps. Mean relative directions of motion of neighbours as a function of the relative coordinates of these neighbours were rendered using MATLAB's *quiver* function. The associated focus $R$ was rendered using the *surf* function.

## 2.2 Collective motion models used to generate synthetic data sets

We applied the alignment-only model of [23], modified to simulate collective motion in bounded and unbounded domains, the zonal model of [20], and the ODE model described in [22] to simulate the trajectories of individuals involved in collective movement in two dimensions. The choice to perform simulations with the alignment-only model in both unbounded and bounded domains was made to try to generate data sets that had some congruence with experimental observations of fish in the wild (with an effectively unbounded domain compared to the limited field of view of a camera; see for example [24]), or aquaria-based observations with clearly defined boundaries (one of the most commonly used systems in controlled experimental studies of collective movement; see for example [10]). However we note that speed moderation and saltatory burst-and-coast motion are characteristic for the movements of many fish, and are not captured by a constant speed model like that of [23]. The simulation data from both the zonal and ODE models was used previously in an investigation of the accuracy of force mapping in extracting repulsion and attraction interactions [21]. The models, and all methods of analysis, were encoded in MATLAB. Unlike the alignment-only and zonal models, the ODE model [22] does account for changes in individual speed in a manner that appears similar to that suggested by force-map analysis of the movements of fish (compare Figure SF2 from [21] with Fig 1C and S12 (middle column) of [2], Fig 2A in [10], Fig 2 in [11] and Fig 6 in [26]). The ODE model also differs from both the alignment-only and zonal models, in that it does not have a prescribed mechanism for orientation interactions. The use of the ODE model [22] here serves as contrast to the other two models; if the force mapping method described in section 2.1.1 captures the prescribed orientation interactions of the zonal model and alignment only model then the same sorts of features should not be captured by the method when applied to data derived from the ODE model [22]. Simulation sets for each model varied across a number of parameters, including group size, $N$, the parameters that control the spatial extent over which particular interaction rules apply, and in some cases were categorised based on emergent motion. Zonal model simulation data was categorised by visual inspection as demonstrating emergent parallel motion, or cohesive motion (a combination of swarming and milling behaviour). ODE simulation data was categorised in greater detail as exhibiting parallel motion, swarming, or milling motion. Milling motion was itself subdivided into sets of clockwise milling, anticlockwise milling or double mills (motion comprised of counter rotating annuli of individuals). For consistency across models, the bulk of the detailed results presented throughout this paper are derived from simulations of groups of $N = 10$ individuals, unless otherwise specified. We provide full details of our implementation of the models, along with relevant exact forms of pairwise interactions for the alignment-only and zonal models, and the particular parameter values used in simulations in the S1 File, section S2 in S1 File. The exact pairwise orientation interactions for the alignment-only and zonal models are used as a point of comparison against interactions inferred via force-mapping.

## 2.3 Estimating the area of regions of high alignment in graphs of local alignment

In the case of data derived from the alignment-only model, initial inspection of plots that illustrated the mean relative directions of motion of neighbours as a function of the relative locations of these neighbours suggested that these plots may capture aspects of the size and geometry of the orientation zone. In particular, a roughly circular region centred on the focal individual, with relatively high values of $R$ (rendered in reddish colours in the plots), approximately scaled in size with increasing orientation/alignment zone radius, $r_o$. We estimated the area of this region as follows, and examined how this correlated with the area of the prescribed

**Table 1. Results of simple linear regression analysis examining the relationship between the areas of the regions of high alignment and the model-prescribed alignment zone with fixed group size (N) in an unbounded domain.** A *p*-value of less than 0.05 means that there is positive linear relationship between the areas of the regions of high alignment and the model-prescribed alignment zone (analysis performed by R-studio).

| Linear regression for area of the regions of high alignment Vs area of the model-prescribed alignment zone | | | | | |
|---|---|---|---|---|---|
| Group Size | R-Squared | p-Value | Slope | y-intercept | F-Statistics |
| 4 | 0.9994 | 5.496e-06 | 1.0178 | -0.3021 | 5438 |
| 6 | 0.9985 | 2.461e-05 | 1.03558 | -0.13562 | 2000 |
| 8 | 0.9993 | 8.619e-06 | 1.08245 | -0.45178 | 4028 |
| 10 | 0.9917 | 0.0003203 | 1.25296 | -1.04370 | 359.5 |
| 12 | 0.9805 | 0.001162 | 1.3690 | -1.5931 | 150.9 |
| 14 | 0.9777 | 0.001424 | 1.4292 | -1.7206 | 131.5 |
| 16 | 0.9738 | 0.001819 | 1.7141 | -2.6091 | 111.3 |
| 32 | 0.9612 | 0.003281 | 3.9329 | -8.8928 | 74.33 |

orientation zone. We first filtered the region where the absolute angular difference between the focal individual and its group mates was less than or equal to 12.5 degrees. Then the *R* values in the filtered region were enclosed with a contour using the MATLAB *contour* function at a contour level of 0.975 (that is, where $R = 0.975$). The contour was then treated approximately as a polygon, described by a discrete, ordered set of vertices with known $(x, y)$ coordinates. We calculated the area of the polygon using the Gauss area formula (also known as the shoelace formula or surveyor's formula; see for example, [29]) for finding the area of the polygon using the coordinates of the vertices. We then compared the estimated area of this zone of greatest alignment with the area of individuals' prescribed orientation zones.

## 3 Results

### 3.1 Results from analysis of simulated data from orientation only model

Here we present results of the analysis of data from simulations with $N = 10$ individuals, which are representative of the sorts of results observed across other group sizes. Elements of the analysis applied for other group sizes are incorporated into summary Fig 6, and Tables 1 to 4.

Force mapping that examined mean changes in direction of motion as a function of neighbour distance and differences in directions of motion correctly identified the qualitative form of the orientation interactions, particularly the sense of rotation required to match directions with neighbours (see the top two rows of Fig 2 for the case of the unbounded domain, and S5 Fig in the S1 File, Section S3.1 S1 File, for the bounded domain). The size of the regions over which force-mapping suggested orientation interactions increased as the model-prescribed radius of orientation, $r_o$, increased. In the case of the unbounded domain, the greatest distance over which orientation interactions appeared to apply more closely approximated the model

**Table 2. Results of simple linear regression analysis examining the relationship between the areas of the regions of high alignment and group size (N) with fixed interaction radius ($r_o$) in an unbounded domain.**

| Linear regression for area of the regions of high alignment Vs Group Size | | | | | |
|---|---|---|---|---|---|
| Interaction Radius | R-Squared | p-Value | Slope | y-intercept | F-Statistics |
| 0.5 | 0.0747 | 0.5124 | 32.99 | -8.724 | 0.4845 |
| 1 | 0.9274 | 0.000122 | 34.966 | -94.27 | 76.66 |
| 1.5 | 0.844 | 0.001263 | 8.676 | -52.198 | 32.46 |
| 2 | 0.8719 | 0.000691 | 0.8947 | -2.432 | 40.84 |
| 2.5 | 0.9493 | 4.158e-05 | 0.49037 | -2.28671 | 112.3 |

**Table 3. Results of simple linear regression analysis examining the relationship between the areas of the regions of high alignment and the model-prescribed alignment zone with fixed group size ($N$) in a bounded domain.**

| Linear regression for area of the regions of high alignment Vs area of the model-prescribed alignment zone | | | | | |
|---|---|---|---|---|---|
| Group Size | R-Squared | p-Value | Slope | y-intercept | F-Statistics |
| 4 | 0.9993 | 8.692e-06 | 0.213713 | 0.526042 | 4005 |
| 6 | 0.991 | 0.0003619 | 0.44106 | 0.44106 | 331.2 |
| 8 | 0.9959 | 0.0001115 | 0.42487 | 0.26887 | 728.9 |
| 10 | 0.9914 | 0.0003378 | 0.34960 | 0.69033 | 346.9 |
| 12 | 0.9882 | 0.0005462 | 0.38661 | 0.28788 | 251.2 |
| 14 | 0.9973 | 5.902e-05 | 0.41379 | 0.24989 | 1115 |
| 16 | 0.9957 | 0.0001204 | 0.54714 | 0.07328 | 692.4 |
| 32 | 0.9835 | 0.0009054 | 0.42077 | 0.41609 | 178.6 |

prescribed $r_o$ than in the case of the bounded domain, but there were often many empty bins over regions where the distance to neighbours $d_{i,j} < r_o$, especially for larger magnitude angular differences (white regions in panels F to J of Fig 2 correspond to empty bins). Simulated groups in the bounded domain tended to become highly compressed, and hence distances closer to focal individuals had better data coverage than in the unbounded case.

Plots for $\frac{\Delta\theta}{\Delta t}(x, y, \varphi_{i,j})$ derived from simulations performed in an unbounded domain in Fig 3, S6 and S7 Figs in S1 File somewhat reveal the qualitative form of the underlying prescribed orientation rule. When the angular difference $\varphi_{i,j} \in (-60°, 60°]$ there are pale blue and yellow dots at the center of the plots (Panels K, L, M and N) which are similar to that in the corresponding plots illustrating the analytical pairwise interactions (Panels E, F, S and T), and are consistent with turns to align with neighbours within those regions; however these regions are difficult to discern against the mostly green background (corresponding to zero turning speed). For angular difference $\varphi_{i,j} \in (-180°, -60°] \cup (60°, 180°]$, in the case where $r_o = 0.5$ (panels G, H, I, J, O, P, Q, and R of S6 Fig in S1 File), there is a small blue dot at the centre of panel G and a small red dot at the centre of panel R of S6 Fig in S1 File; this is similar to the corresponding plots illustrating the analytical pairwise interaction (Panels A and X), and is consistent with turns to align with neighbours within those regions. Panels (H, I, J, O, P and Q) in S6 Fig in S1 File do not clearly indicate an individual's tendency to turn to align with neighbours within those regions. In Fig 3 and S7 Fig in S1 File (panels G, H, I, J, O, P, Q, and R), where angular difference $\varphi_{i,j} \in (-180°, -60°] \cup (60°, 180°]$, corresponding plots are difficult to interpret due to a lack of data/empty bins close to the focal individual. Fig 4, S8 and S9 Figs in S1 File contain approximately circular red and blue regions centred on the focal individual which are consistent with turns to align with neighbours in these regions. The tendency

**Table 4. Results of R-Squared values for the relationship between the estimated area of the alignment zone and group size ($N$) with fixed interaction radius ($r_o$) in a bounded domain.** The R-Squared values here are very low, with the greatest (0.2104) accounting for only 21.04% of the observed variance. Hence linear regression is not performed for this particular set of analysis.

| Interaction Radius | R-Squared |
|---|---|
| 0.5 | 0.01982 |
| 1 | 0.2104 |
| 1.5 | 0.004692 |
| 2 | 0.1684 |
| 2.5 | 6.889e-05 |

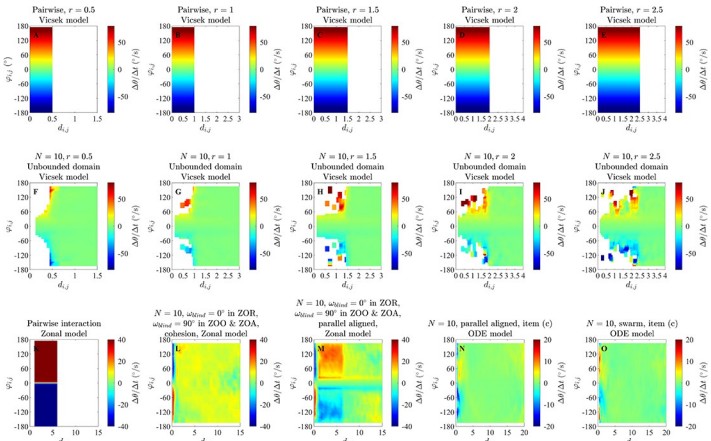

**Fig 2.** Changes in direction of motion, $\Delta\theta/\Delta t$ (in degrees per second), as a function of the distance to a neighbour, $d_{i,j}$ (in unspecified spatial units), and angular difference in directions of motion, $\varphi_{i,j}$ (in degrees), for exact pairwise interactions from the alignment only (panels A to E) and zonal (panel K) models, and as inferred via force mapping for simulations of the alignment only model in an unbounded domain (panels F to J), zonal model (panels L and M) and ODE model (panels N and O). For all plots, positive turns are anticlockwise (redder regions) and negative turns are clockwise (bluer regions). Panels A to E illustrate the exact turning response of a focal individual to a single neighbour for interaction/orientation zones of increasing radius for the alignment-only model (see equation (S2.9) and its derivation in Section S2.1.1 of the S1 File); panels F to J illustrate force maps for groups of $N = 10$ individuals in an unbounded domain for the same increasing sequence of interaction radii. White regions in panels A to E correspond to regions where no interaction applies, whereas white regions in panels F to J coincide with empty bins/no data. Panel K illustrates the exact turning response of an individual to a single neighbour in its zone of orientation for the zonal model, with a repulsion zone radius $r_r = 1$, orientation zone width $\Delta r_o = 5$ and attraction zone width $\Delta r_a = 8$ (such that the orientation zone extends between 1 and 6 units from a focal individual, and the attraction zone extends from 6 to 14 units; see equation (S2.21), SI Section S2.2.3). White regions in panel K correspond to regions where no orientation interaction applies. Panels L and M show force maps for simulations of the zonal model with groups of $N = 10$ individuals for the same zone sizes as panel K, for groups that exhibited cohesive but non-parallel motion (panel L) or parallel motion (panel M). The particular simulations used to generate panels L and M also applied a blind zone that extended across the zones of orientation and attraction (but not repulsion). Panels N and O illustrate force map inferred turning responses from simulations of the ODE model with $N = 10$ individuals where parallel motion (panel N) or swarming emerged.

to align is clearer when the angular difference is greater in the case of the bounded domain. The spatial extent of the region over which alignment appears to occur seems to be slightly larger than the model prescribed region in the bounded domain case.

Plots for the mean relative directions of motion of neighbours as a function of relative neighbour locations in Fig 5 suggest that there may be a correlation between the size of the red region and the prescribed orientation zone. This region of high alignment is the deepest red region extending around the focal individual. We determined the area of the estimated region of greatest alignment using the method noted in section 2.3 and present the results in the line graphs in Fig 6 and Tables 1 to 4. It is evident from the line graph plots in panels A and B of Fig 6 that as the model interaction radius $r_o$ increased then there was an increase in the radius of the approximately circular red region in the local alignment plot. This result was consistent for calculations carried out in unbounded and bounded domains. Given the general relationship between the radius of a circle, $r$, and the area of the same circle, $\pi r^2$, it is reasonable to expect that the relationship between the prescribed alignment radius, $r_o$, and the area of the region of greatest alignment might be close to quadratic, and visually that seems to be the case here (or at least the area of the region increases more rapidly than linear growth as the radius increases).

We applied linear regression analysis to examine the relationship between the areas of the regions of high alignment and the model-prescribed alignment zone (data illustrated in panels

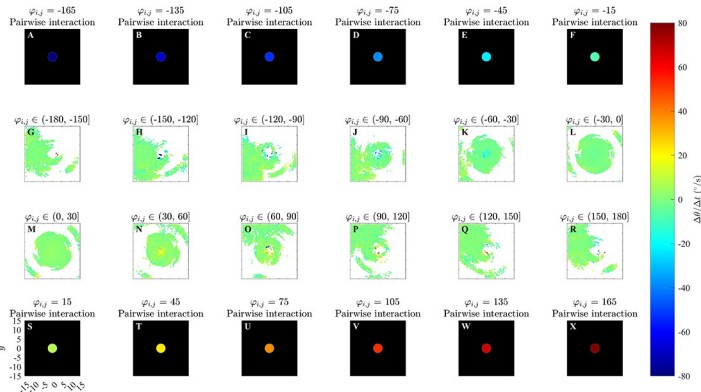

**Fig 3. Force maps of Δθ/Δt and associated exact turning responses to a single neighbour as a function of the relative (x, y) coordinates of neighbours and angular difference in directions of motion, $\varphi_{i,j}$, for the alignment only model in an unbounded domain.** Panels A-F and S-X: analytical pairwise interactions, where turning of the individual is governed by equation (S2.9). Panels G-R illustrates changes in direction of the focal individual as a function of positions of neighbours and the differences in directions of motion between a focal individual and these neighbours via force mapping analysis of simulations from the alignment only model with $r_o = 2.5$ and $N = 10$. The focal individual is located at the origin and moving right, parallel to the positive x-axis. Positive turns are anticlockwise and negative turns are clockwise. White regions indicate that no partner individuals were recorded with the corresponding relative (x, y) coordinates. For panels A-F and S-X, the black region indicates no interactions. $\varphi_{i,j}$ is measured in degrees here.

E and F of Fig 6; regression details in Tables 1 and 3). The analysis suggested relatively high positive correlation between these two variables for both unbounded and bounded domains and across all simulated group sizes. Corresponding R-Squared values ranged from 0.9612 to 0.9994 (groups of $N = 3$ 2 and $N = 4$ respectively in an unbounded domain) and from 0.9835 to 0.9993 (groups of $N = 32$ and $N = 4$ respectively in a bounded domain). Associated lines of best fit had slopes that all differed significantly from zero with a standard threshold for significance of 0.05.

In an unbounded domain, there was no correlation between group size and the area of the region of high alignment when $r_o = 0.5$ (Table 2). However, there was quite strong positive correlation between these measures for $r_o = 1, 1.5, 2$ and $2.5$, and associated regression lines in these cases had slopes that differed significantly from zero (Table 2). For interaction radii of $r_o$

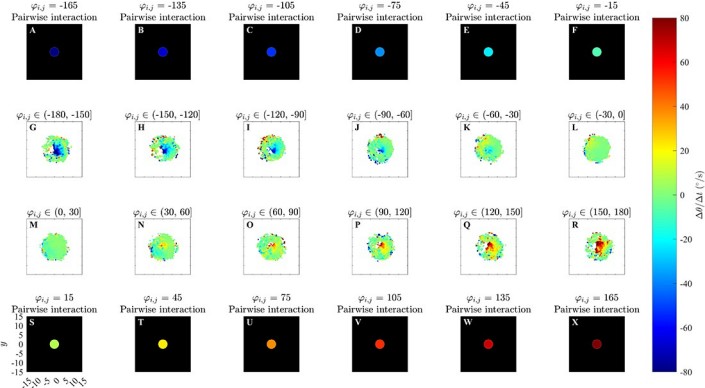

**Fig 4. Analytical pairwise interactions and results of force mapping analysis of simulated data from the alignment only model with $r_o = 2.5$ and $N = 10$ in a bounded domain.** All other details are the same as for Fig 3.

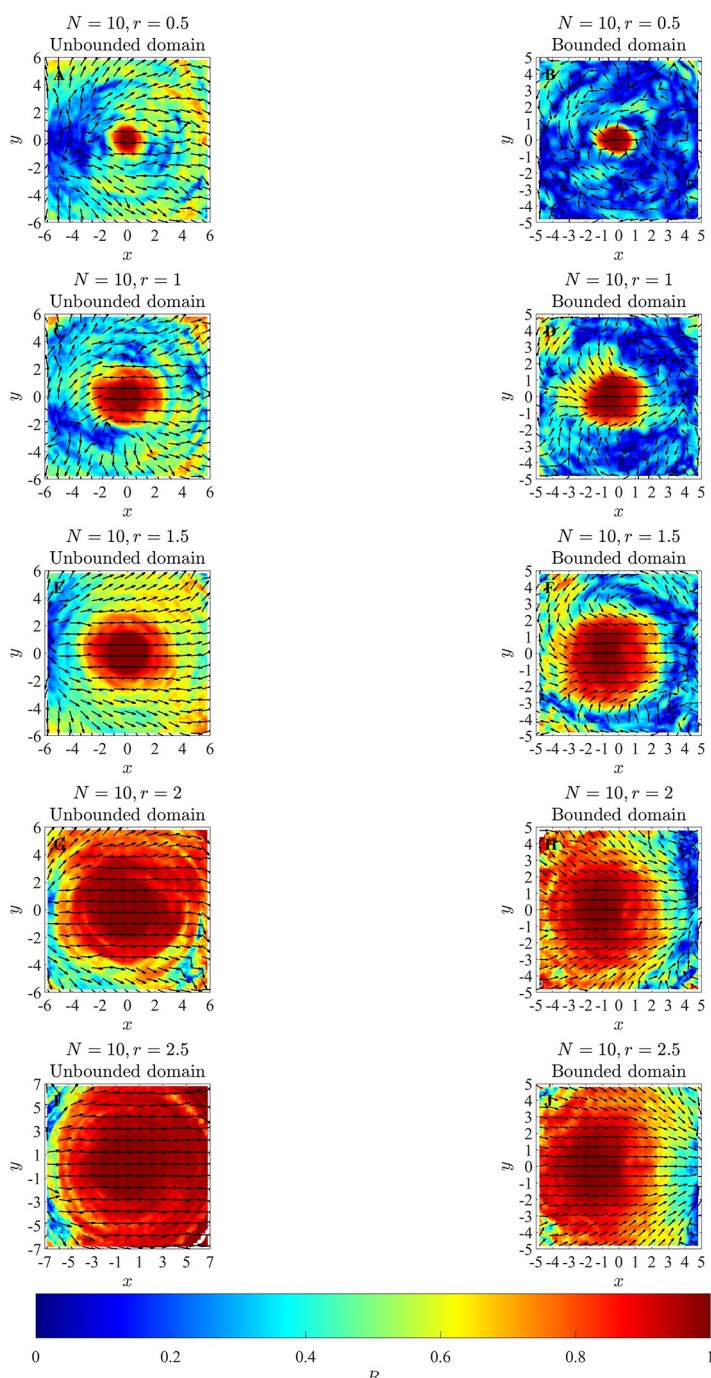

**Fig 5. Left and right columns illustrate mean relative directions of motion of neighbours as a function of relative group mate locations, as observed in an unbounded and a bounded domain respectively.** In these plots, the focal individual is located at the origin and moving right parallel to the positive $x$-axis. Arrows point in the mean relative direction of motion of all other individuals relative to the focal individual. Colours represent differing values of $R$, which describes the focus of the angles in a given bin about the mean.

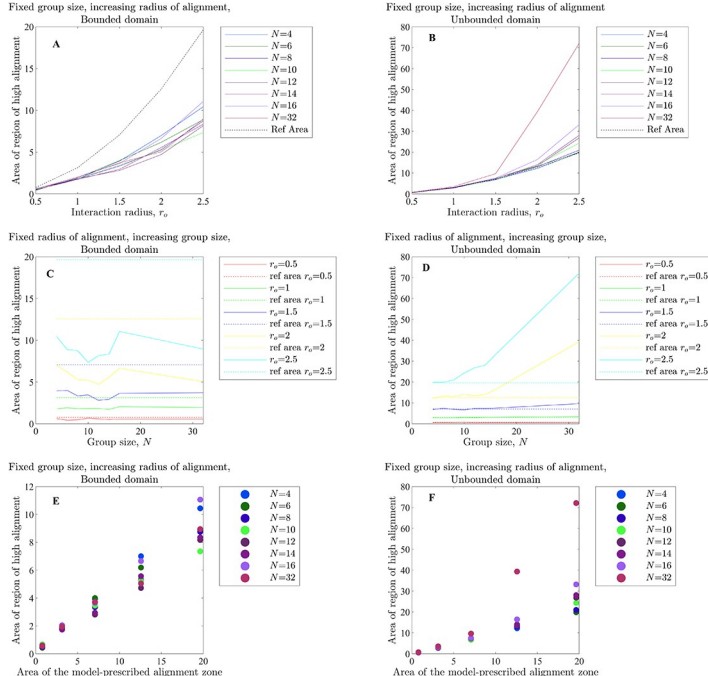

**Fig 6. The relationship between regions of high alignment (the red central region of the local alignment plots) and prescribed alignment interaction radii, $r_o$, (top row), group size, $N$, (middle row) and area of the model prescribed alignment zone (bottom row) for simulations carried out in a bounded domain (left column) and an unbounded domain (right column).** Given the circular nature of the prescribed alignment zone, and the relationship between the radius, $r$, and area of a circle, $\pi r^2$, it is reasonable to expect an approximately quadratic relationship between the area of the region of high alignment and $r_o$ in A and B. As a consequence, we did not perform linear regression on the data presented in A and B. In C and D, dashed lines illustrate the area of model prescribed interaction zones for reference. Linear regression analysis was performed on the data presented in panels D to F, with further details provided in the main text, and Tables 1 to 4.

= 1, 1.5, 2 and 2.5 units, the area of the region of greatest alignment on local alignment plots tended to be greater than the area of the prescribed zone of alignment (panel D of Fig 6).

The areas of regions of high alignment were not correlated with group size in a bounded domain (Table 4). The areas of regions of high alignment were also systematically smaller than the area of the alignment zone in all instances in a bounded domain (panel C of Fig 6).

## 3.2 Results from analysis of simulated data from zonal model

Here, we focus on results for simulations of the zonal model with $N = 10$ individuals, repulsion zone radius, $r_r = 1$, orientation zone width $\Delta r_o = 5$ and attraction zone width $\Delta r_a = 8$ (see the S1 File, Section S2.2 in S1 File for more details on these parameters). This data subset includes instances where individuals' blind zones extended across all of the zones of repulsion, orientation and attraction, just the zones of orientation and attraction, or did not exist. In cases where there was a blind zone, emergent states included both parallel group motion and cohesive groups that did not move in parallel. The data subset also includes sets of simulations run for durations of 1000 and 10000 time steps. For other combinations of model parameters listed in Table 2 in [21] (reproduced in S4 Table of the S1 File here), we obtained similar observations, however we do include some of these where differences were apparent.

In plots of $\frac{\Delta \theta}{\Delta t}(d_{i,j}, \varphi_{i,j})$ for the groups that formed parallel aligned motion individuals were identified as turning to align with their neighbours most rapidly from $d_{i,j} = 1$ to $d_{i,j} = 6$ (Fig

2M, and S10B, S10D and S10F Fig in S1 File), although the analysis suggests a tendency to align outside this region with slower turns as well. This region where $1 \leq d_{i,j} \leq 6$ is the model prescribed zone of orientation in the underlying simulations. The average turning speed identified by force mapping was systematically lower than that prescribed for pairwise interactions (for example, compare the prescribed orientation rule for pairs in Fig 2K with the data derived response identified in Fig 2M, as well as S10A Fig in S1 File, with panels B, D and F of S10 Fig in the S1 File). On the other hand, for cohesive groups (Fig 2L, and S10C and S10E Fig in the S1 File) such tendency to align was not evident. For both forms of emergent motion, individuals tended to turn away from the direction of motion of very near neighbours (those with $d_{i,j} <$ 1, that is, within the model-prescribed zone of repulsion) (see Fig 2, panels L and M).

In the plots for $\frac{\Delta\theta}{\Delta t}(x, y, \varphi_{i,j})$ in Figs 7 and 8 and S11 to S14 Figs of the S1 File, it can be observed that for differences in direction in the interval $\varphi_{i,j} \in (-30\degree, 0\degree]$ for groups exhibiting parallel aligned motion, there is a bluer circular region surrounding the repulsion region discussed in Mudaliar et al. [21] (Fig 7R, S11L and S13R Figs in S1 File) and a redder circular region surrounding the repulsion region when $\varphi_{i,j} \in (0\degree, 30\degree]$ (Fig 8C and S12B and S14C Figs in S1 File) which suggest a tendency for the focal individual to align with neighbours at shorter ranges, consistent with the underlying model rule. This qualitative consistency between model-prescribed interactions, and those inferred via force mapping when differences in heading are relatively small, and group motion is parallel, extends to both short range repulsion behaviour and longer range attraction behaviour (compare panels P and R in Fig 7 and S13 Fig in S1 File, panel A and C in Fig 8 and S11 and panel A and B in Fig S12, S14, S14K and S14L Figs in S1 File). In addition, the presence of the prescribed blind zone may be faintly evident in some plots of $\frac{\Delta\theta}{\Delta t}(x, y, \varphi_{i,j})$, including Fig 7R, where there is a sector trailing the focal individual where turning responses were closer to zero in magnitude, and a similar region appears in Fig 8C. The plots for $\frac{\Delta\theta}{\Delta t}(x, y, \varphi_{i,j})$ for groups that exhibited cohesion without parallel motion do not show the same tendency to align for any range of differences in headings. There were some other consistent patterns in the case of cohesive motion that included a tendency to turn clockwise when partners were in front and anticlockwise when partners were behind for small to moderate magnitude negative differences in heading (see for example Fig 7 and S13 Fig, panels N and Q in S1 File), with this pattern reversed for positive differences in heading (see for example Fig 8 and S14 Fig in S1 File, panels B and E). These patterns are interesting, but they do not reflect the underlying model prescribed behavioural rule for orientation with neighbours at distances from 1 to 6 units. We compared plots for $\frac{\Delta\theta}{\Delta t}(x, y, \varphi_{i,j})$ obtained from the simulated data for simulations run for 1000 and 10000 timestep (S15 to S20 Figs in the S1 File) for the case where the emergent pattern was parallel aligned motion. We found that when the simulation was run for 10000 time steps there was not much difference compared to the simulation run for 1000 time steps. As with the analysis in [21], the force maps examined here also suggested turning behaviour consistent with model-prescribed avoidance of neighbours at short range, and model-prescribed attraction to neighbours at longer range.

In plots of the mean relative directions of motion of neighbours as a function of relative neighbour positions (Fig 9A, 9C, 9E, 9G and 9I) for groups that formed parallel aligned motion, a red region surrounds the focal individual, comprised of a roughly annular or elliptical region centred on the focal individual, and two extended regions to the front and back of the focal individual. The annular/elliptical part of the red region is of similar extent to the zone of orientation that is prescribed by the model, which extends from 1 unit to 6 units distance from the focal individual. The red region to the front and behind the focal individual reflects that the binned angles indicate largely similar directions. For some sets of groups that formed cohesive patterns of movement, other than parallel motion, such as swarming or milling,

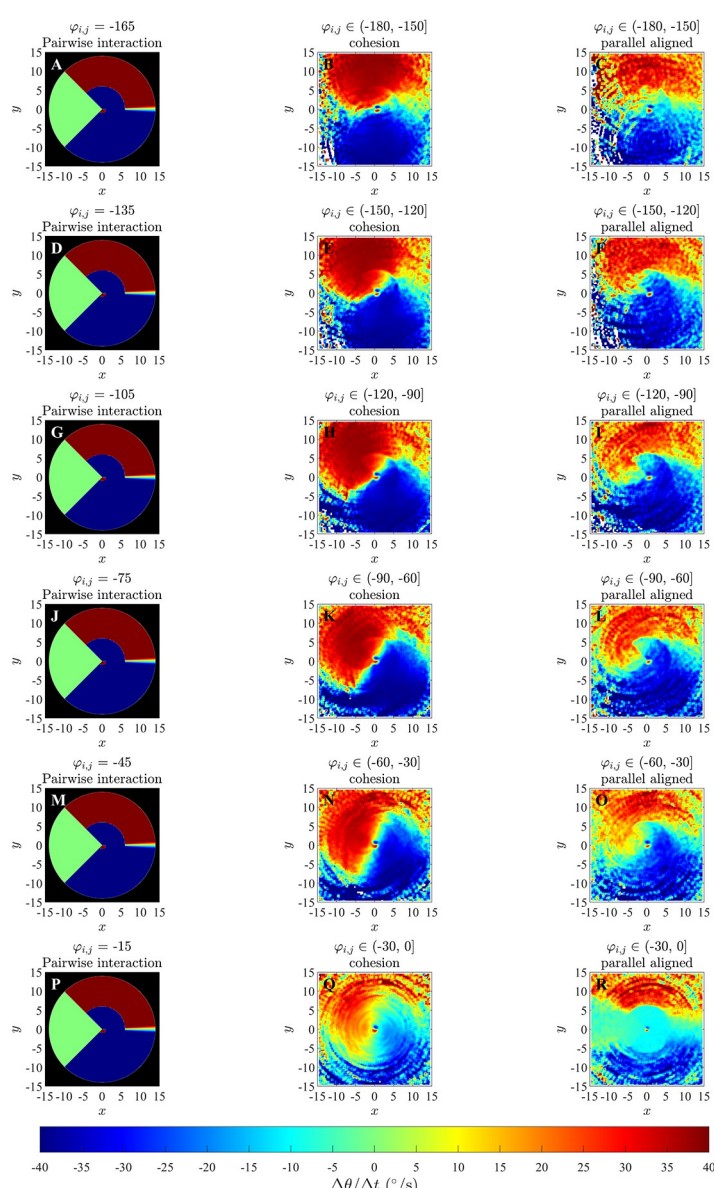

**Fig 7.** Left column: exact analytical pairwise changes in direction as prescribed by the zonal model (SI Section S2.2 in S1 File) with $r_r = 1; \Delta r_o = 5; \Delta r_a = 8; \varphi_{i,j} \leq 0$ and $\omega_{blind} = 90°$, where angular differences are negative. The focal individual will not adjust its velocity in response to neighbours in the black region (exterior to the zone of attraction), nor to neighbours in the blind zone (here, the green coloured wedge trailing the focal individual). Middle and right column: results of force mapping where emergent motion was cohesive, but not parallel (middle column), or parallel (right column).

redder regions are apparent to the left and right of the focal individual at the origin, but with substantially lower focus (*R*-value) than for parallel motion (Fig 9B and 9D). Unlike the parallel motion cases, the mean directions of motion of partners over the regions of greatest focus were not closely matched with that of the focal individual. Instead, the patterns in these graphs suggest consistent circulation of group mates in either anticlockwise directions (to the left of the focal individual) or clockwise directions (to the right of the focal individual). For some zone sizes such as where $r_r = 1.5$, $\Delta r_o = 1.5$ and $\Delta r_a = 11$ (Fig 9F, 9H and 9J) with cohesive, but

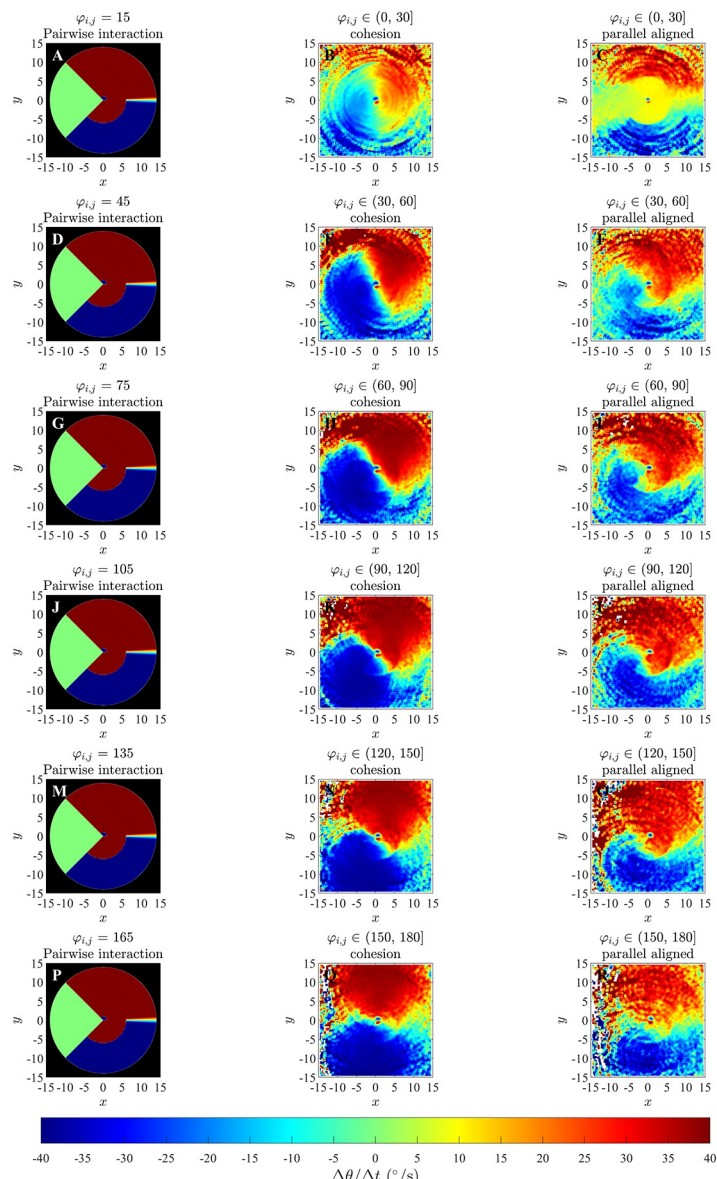

**Fig 8.** Left column: analytical pairwise changes in direction as prescribed by the zonal model (SI Section S2.2 in S1 File) with $r_r = 1; \Delta r_o = 5; \Delta r_a = 8; \varphi_{i,j} > 0$ and $\omega_{blind} = 90°$, where angular differences are positive. Middle and right column: results of force mapping for emergent cohesive and parallel motion.

non-parallel, motion there was less focus in each set of binned angles about the mean in the structured regions equivalent to the redder regions in Fig 9B and 9D.

## 3.3 Results from analysis of simulated data from ODE model

In plots of $\frac{\Delta\theta}{\Delta t}(d_{i,j}, \varphi_{i,j})$ in the case of emergent parallel motion (Fig 2N, S21E and S22E Figs in S1 File) there may be some indication of a weak tendency to align with neighbours between distances of 1 to 3 units; however, this is not the same clearly defined tendency to align seen in analysis of the zonal model.

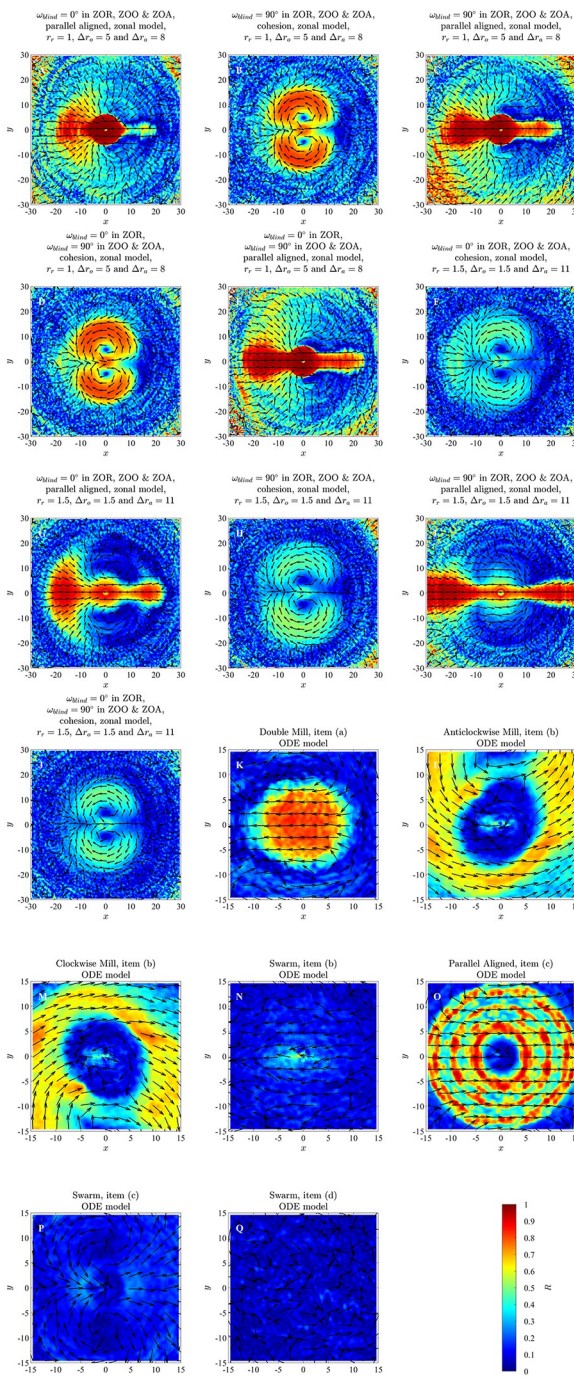

**Fig 9.** Local alignment plots derived from simulations of the zonal model (panels A to J) and ODE model (panels K to Q). Panels A to E are derived from zonal model simulations with $r_r = 1$, $\Delta r_o = 5$ and $\Delta r_a = 8$, whereas panels F to J were derived from plots for simulations with $r_r = 1.5$, $\Delta r_o = 1.5$ and $\Delta r_a = 11$. Panels K to Q were derived from ODE model simulations with various within model parameters, as detailed in S5 Table of the S1 File. In each plot the focal individual is located at the origin, moving parallel to the positive $x$-axis. The arrows point in the mean direction of motion of all other individuals relative to the focal individual. Colours represent $R$, a measure of the focus of binned angles/directions about the mean, with high values of $R$ indicating a high level of agreement with the mean, and lower values corresponding to a greater scatter of angles.

Plots of $\frac{\Delta\theta}{\Delta t}(x, y, \varphi_{i,j})$ did not in general suggest any tendency to align with neighbours in the same way that the same plots suggested such behaviour in some cases for the alignment-only [23] and zonal [20] models (Figs 10 to 12). As noted in [21], in the ODE model $\frac{\Delta\theta}{\Delta t}$ is dependent on the speed $s$ of individuals. However, we did not plot $\frac{\Delta\theta}{\Delta t}(x, y, \varphi_{i,j}, s)$ for the analysis presented because of the narrow range of individual speeds $s$ observed for the analysis of the same data in [21] (Figs 8 and 9).

There were some differences for the local alignment plots compared to similar plots from the alignment only model [23] and zonal model [20]. When the emergent pattern was a double mill, for regions with high $R$ values, the neighbours were travelling in the opposite direction to the focal individual on average (Fig 9K). For the milling cases (Fig 9L and 9M) the arrows point in the respective direction based on whether the mill was a clockwise or anticlockwise mill. There was little or no alignment suggested by the plots for swarms (Panels N, P and Q, Fig 9). Annular regions of relatively strong alignment were apparent when parallel aligned motion emerged, somewhat similar to the central portion of the local alignment plots (Fig 9O) for the zonal model.

## 4 Discussion

Force maps as a function of $(d_{i,j}, \varphi_{i,j})$ are capable of resolving alignment/orientation zones (at least when such zones are symmetric about the origin, as is the case for the simulations examined here (Fig 2F to 2J and 2M, S5, S10B, 10D and 10F Figs in S1 File)), and do not suggest the presence of such zones in the absence of explicit alignment/orientation interactions, or at least not strongly so, even when the associated emergent behaviour is parallel motion (see Fig 2N, and S21C Fig in S1 File, obtained from simulations of the ODE model). However, the strength of the tendency to align inferred by force mapping could still be affected by emergent behaviour (as was the case for cohesive motion of the zonal model [20] seen in Fig 2L and S10 Fig in S1 File). The force mapping derived region over which individuals aligned with their partners tended to be larger than the model-prescribed orientation region when the independent variables of choice were $d_{i,j}$ and $\varphi_{i,j}$. This may have been due to indirect interactions between individuals separated by more than an orientation radius via neighbours common to some individuals. A broad summary of our observations relating to force maps of $\frac{\Delta\theta}{\Delta t}\left(d_{i,j}, \varphi_{i,j}\right)$ for each model and simulation set is provided in Table 5.

The force maps that we examined with greater spatial detail (that is, those where the independent variables where $(x, y, \varphi_{i,j})$) did also resolve orientation zones in some cases, but the conditions under which this resolution was possible were dependent on several factors; see Table 6. For example, an approximately circular region centred on the focal individual, where individuals turned to match the directions of motion of neighbours and of similar size to the model-prescribed interaction zone, was apparent (although barely so over angular difference ranges closer to zero) for cases where differences in heading between focal individual and neighbour were relatively small for simulation data from the alignment-only model in an unbounded domain (Fig 3). The turning response itself became clearer as angular differences increased in the unbounded domain case, but lack of data close to the focal individual meant that it was not really possible to interpret results for the largest magnitude angular differences. The tendency to align was clearer in a bounded domain, especially when differences in heading/angular differences were large (Fig 4, S8 and S9 Figs in S1 File). A tendency to align over a region of similar extent to the prescribed orientation zone of the zonal model was also apparent when the emergent group level motion was parallel, and angular differences were relatively small. However, the tendency to align was at best not clear when angular differences were

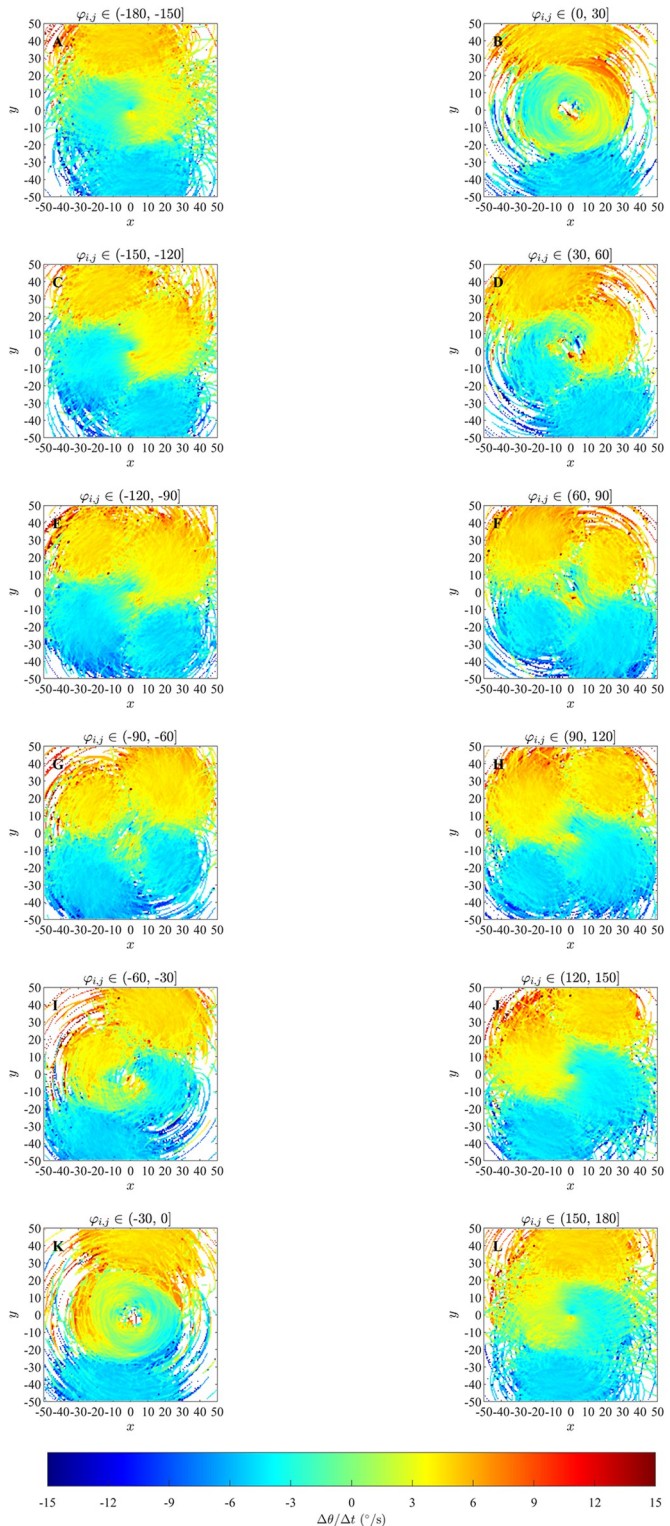

**Fig 10.** Force maps of $\Delta\theta/\Delta t$ as a function of $(x, y, \varphi_{i,j})$ derived from simulations of the ODE model that generated double milling motion; corresponding model parameter values are provided in S5 Table of the S1 File, item (a). In each plot the focal individual is located at the origin, moving parallel to the positive $x$-axis. Anti-clockwise turns are positive, clockwise turns are negative. Left column: $\varphi_{i,j} < 0$, hence neighbour's direction of travel is clockwise relative to the focal individual. Right Column: $\varphi_{i,j} > 0$, hence neighbour's direction of travel is anticlockwise relative to the focal individual.

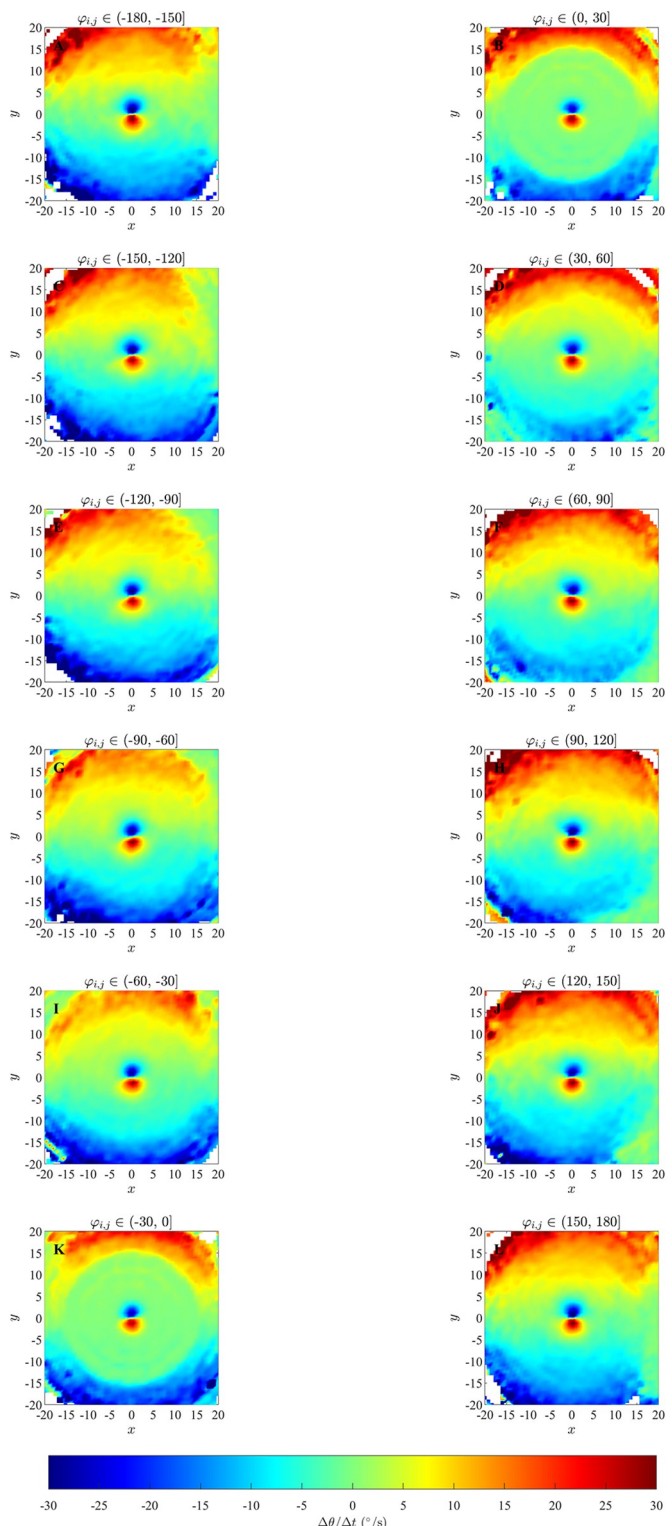

**Fig 11. Force maps of $\Delta\theta/\Delta t$ as a function of $(x, y, \varphi_{i,j})$ derived from simulations of the ODE model that generated parallel motion; corresponding model parameter values are provided in S5 Table of the** S1 File**, item (c).** Details are otherwise the same as for Fig 10.

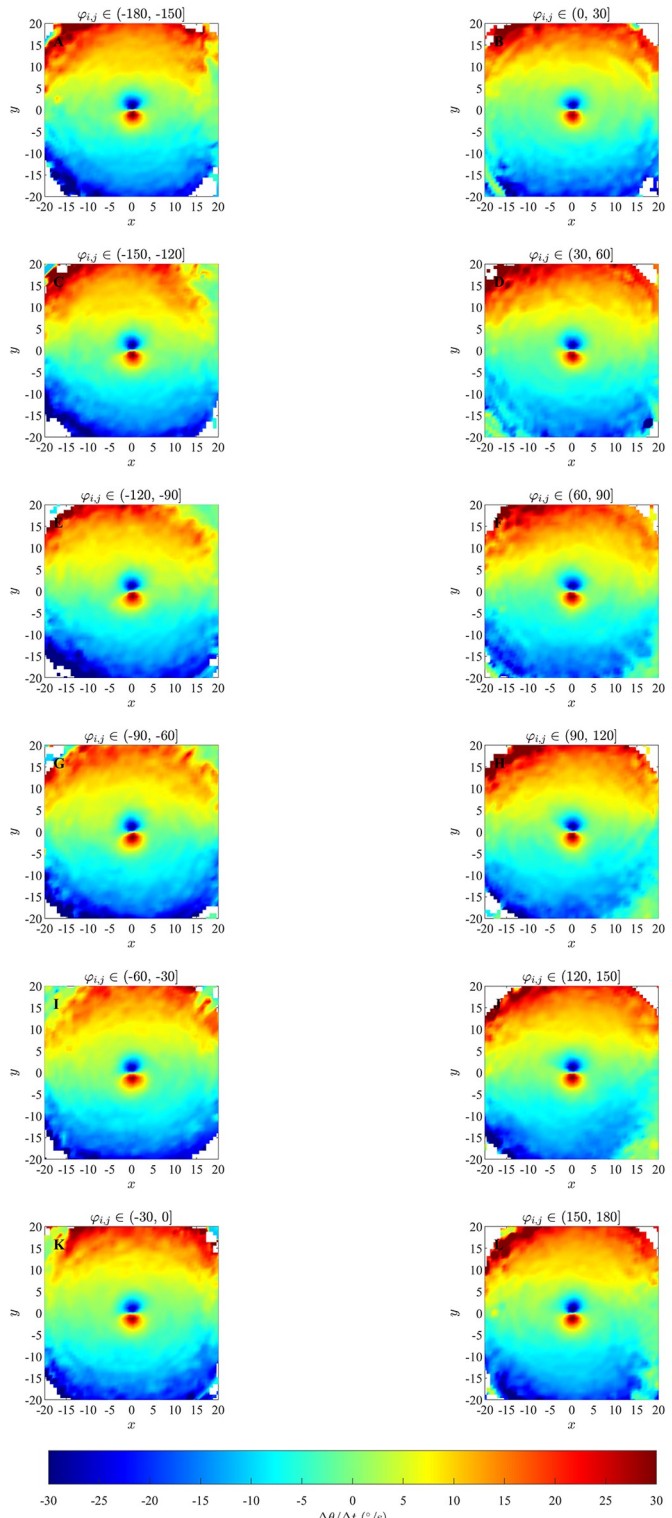

**Fig 12. Force maps of Δθ/Δt as a function of (x, y, φ_{i,j}) derived from simulations of the ODE model that generated swarming motion; corresponding model parameter values are provided in S5 Table of the S1 File, item (c).** Details are otherwise the same as for Fig 10.

**Table 5. Broad summary of force mapping effectiveness for determining $\frac{\Delta\theta}{\Delta t}\left(d_{i,j}, \varphi_{i,j}\right)$.**

| Data Set | Features Identified | Inaccuracies and Other Issues |
|---|---|---|
| Alignment-only model, unbounded domain. | Correct extent and form of turning response. | Many empty bins in regions of interest. Magnitude of turns underestimated. |
| Alignment-only model, bounded domain. | Clear evidence of alignment. No empty bins. | Overestimated distance of alignment interaction. Magnitude of turns underestimated. |
| Zonal model, parallel motion. | Clear evidence of alignment (greatest response in prescibed orientation zone). | Turning response evident outside orientation zone. Magnitude of turns underestimated. |
| Zonal model, cohesive motion. | Perhaps some evidence of alignment, for $1 < d_{i,j} < 2$. | Generally unclear that there is an alignment response. |
| ODE model (no alignment rule). | Unclear or weak alignment suggested at most. | Signs of $\frac{\Delta\theta}{\Delta t}$ consistent with alignment for $d_{i,j} < 4$ for item (c) swarms and parallel groups. |

large when the emergent motion was parallel, and not apparent at all in the case of emergent cohesive non-parallel motion, even though the underlying behavioural rules were the same for both the parallel and cohesive motion cases. Some of these issues might be due to force mapping not including any explicit mechanisms for separating/disentangling combined interactions (either interactions that apply over the same spatial region, or that are combined due to interactions with multiple neighbours), as is the case for orientation and attraction interactions in the zonal model [20]; this was identified as a potential issue by Escobedo et al [9]. The plots of this form also tended to be quite noisy, which makes interpretation even more difficult.

The independent variables chosen for the set of force mapping analyses in this paper were informed by direct knowledge of model rules for orientation/alignment behaviour. The orientation rule applied within the alignment-only model is explicitly a function of $d_{i,j}$ and $\varphi_{i,j}$ (which could also be viewed as a function of $(x, y, \varphi_{i,j})$), whereas the blind zone of the zonal model means that the orientation rule can be written as a function of $(x, y, \varphi_{i,j})$. The simpler pairing of $d_{i,j}$ and $\varphi_{i,j}$ as independent variables was quite reliable in revealing underlying orientation behaviour for the analyses performed here, whereas analysis with the independent variables $(x, y, \varphi_{i,j})$ was less so. Broadly, based on the analysis presented here, it appears that force mapping does not always reveal underlying behavioural rules, even when both the dependent

**Table 6. Broad summary of force mapping effectiveness for determining $\frac{\Delta\theta}{\Delta t}\left(x, y, \varphi_{i,j}\right)$.**

| Data Set | Features Identified | Inaccuracies and Other Issues |
|---|---|---|
| Alignment-only model, unbounded domain. | Evidence of alignment with magnitude and over distance of prescribed rule. | Many empty bins, including over alignment zone. |
| Alignment-only model, bounded domain. | Clear evidence of alignment, especially for large magnitude angular differences. | Distortion of size and shape of orientation zone. Many empty bins. |
| Zonal model, parallel motion. | Alignment evident for small magnitude angular differences. Blind zone faintly visible when present. | Beyond angular differences of magnitude 30°, alignment behaviour is not evident. Incorrect magnitude of turning response (when visible). |
| Zonal model, cohesive motion | | Prescribed alignment rule not evident in force maps. |
| ODE model (no alignment rule). | No tendency to align suggested, consistent with model rules. | |

variable (here $\Delta\theta/\Delta t$) and the independent variables are chosen to match exactly the variables that define that behavioural rule.

Examination of the local alignment plots (Fig 5) derived from alignment only model [23] data suggested that there is a correlation between the sizes of the regions of greatest focus (and mean directions similar to the focal individual) and the area of the prescribed interaction zone. However, the pattern is not as clear with the zonal model [20], being strongest (but differing in detail) when there is parallel motion (Fig 9A, 9C, 9E and 9G), and differing quite a lot when the emergent motion is one of cohesion (Fig 9B, 9D, 9F, 9H and 9J). Further different emergent-state dependent patterns are evident in plots (Fig 9K to 9Q) from the ODE model [22], but an annular region with high alignment between focal individual and neighbours and relatively high $R$ values appears when the emergent motion is parallel (Fig 9O). This pattern for the ODE model [22] emerges even though there is no explicit alignment mechanism in the model. We therefore think that although we observed correlations with the alignment only model [23], broadly the local alignment plots are quite strongly affected by emergent behaviour, rather than the details of the orientation rule/interaction. Generally the plots seem to be more indicative of emergent behaviour, rather than the underlying interactions driving this behaviour.

Based on the work here, it seems that with suitably chosen independent variables force maps can be a useful tool for identifying the presence of explicit alignment interactions. For the exploration described here, the clearest evidence for alignment behaviour was elucidated from force maps that examined changes in direction of motion as a function of the distance to neighbours and differences in direction of motion. This choice of variables for the resulting force maps was directly informed by knowledge of the form of alignment rules in the models that we used; it is not necessarily the case that these would be the best variables to use when analysing movement data from real animals, but it may be a reasonable starting place. With the ability of force maps to reveal orientation behaviour, and the benefit of the relative simplicity of this approach come some inaccuracies, including those identified in previous work. These inaccuracies include the ability to accurately identify the spatial extent of an orientation rule, which was often overestimated by the analysis presented here, the effects of differing patterns of group motion on the force maps, as had previously been identified in [21], and in some cases incomplete coverage of regions of interest by the maps due to insufficient data, an issue discussed in [9].

Future work could seek to address at least some of the inaccuracies and issues with force mapping identified both here, and in [9, 21]. One such shortcoming is that of incomplete coverage of regions of interest associated with empty bins. Such a problem is not unique to force mapping, and can be an issue for any numerical method applied to fit a function to data. We think the problem of incomplete coverage can be exacerbated by increasing the number of independent variables under consideration. A straightforward way to try to address incomplete coverage is by simply supplying more data for the analysis, but care needs to be taken in how additional data is supplied. S15 to S20 Figs in S1 File compare force maps derived from equal numbers of simulations from the zonal model which result in emergent parallel motion, for durations of 1000 and 10000 time steps. Incomplete coverage, illustrated as white regions in the plots, is evident in the force maps for both durations, over similar regions of $(x, y, \varphi_{i,j})$ space, especially for larger magnitude values of the angular difference, $\varphi_{i,j}$. We think this is likely due to the fact that parallel aligned motion tends to emerge very quickly for the zonal model, within the order of several hundred time steps, and thereafter individuals tend to maintain relatively stable positions within the group. Due to these relatively stable positions and the stable direction of motion of the group and individuals, the individuals then only provide data for a subset of the bins in the domain, and adding longer durations of observations/data will not necessarily fix this problem. An alternative method to try to address the problem of

incomplete coverage could be to increase the number of independent simulations or observations, rather than the durations of the observations. However, we note that the data sets derived from the zonal model and presented here were comprised of 80 different simulations each, substantially more than what is usually provided by a real world experimental study, and suggesting that even large numbers of independent observation sets may not prevent issues with empty bins. A further consideration relating to the data used throughout this study is that each of the models frequently generates relatively stable emergent motion that persists for thousands of time steps, whereas real animals, such as fish and humans, engage in collective motion at times that frequently transitions from one emergent pattern to another over durations of tens to hundreds of seconds [30, 31]. We speculate that force maps generated from data where groups do not stay in a particular configuration for extended periods may not be affected as much by incomplete coverage.

A common drawback to force mapping revealed both here and in previous work in [21] is the clear effect that emergent patterns of motion can have on the maps. As noted in [21], Attanasi et al. [32] applied a series of corrections to analysis of group level measures of motion to account for factors such as consistent rotation about the group centre, and the periodic expansion and contraction of the group. It would be of immediate interest to apply these documented corrections within force mapping calculations, to see if they can mitigate emergent group level behaviour from affecting force maps.

Both this work, and previous work in [21], has focussed on interactions that apply over metric (real-world distance) based scales. However, a number of empirical studies have pointed to the potential importance of interactions that may apply over topological scales (measured via distance-based neighbour rank, rather than linear distance) [33], interactions that are dominated by nearest and most influential neighbours [34], the visual networks of individuals [35, 36], and locomotory behaviour other than just averages of velocity changes, such as the burst and coast behaviour exhibited by many shoaling fish [7]. It would be of interest explicitly interrogate the usefulness and accuracy of force-maps in some of these scenarious in future work.

Finally, and importantly, is consideration for how the work presented here might affect or inform future analyses of real world data. In the first instance, we have demonstrated that force mapping can be targeted to alignment interactions of the form hypothesised for a number of theoretical models, and that such force maps can and do reveal the presence of alignment interactions in many cases. As noted in the introduction, the use of force maps to investigate alignment interactions has been quite limited, especially when compared to the overall number of studies that have used force mapping techniques. Both [10] and [17] used force maps to examine the turning behaviour of fish in response to angular differences with neighbours, and the directions to these neighbours. In [10], such analysis seemed to suggest that the study species for that work, eastern mosquitofish (*Gambusia holbrooki*), were not actively aligning with each other via direct interactions. However, the choice of independent variables in [10] may not have been the best or most direct for identifying alignment interactions, as compared to the variables chosen in our study here. Further, another more recent study on eastern mosquitofish suggests these fish do align with each other, based on analysis of correlations in direction of motion over a range of time lags [26]. Hence, it could be very worthwhile to perform force mapping analysis of the interactions of mosquitofish that examines changes in direction of motion as a function of neighbour distances and angular differences, as has been shown to work here, to try to further investigate and reconcile if this species does apply an explicit alignment rule in interaction with neighbours. Other species previously studied in detail, such as those in [2, 11, 17], could also be revisited to gain a fuller picture of the underlying interaction rules that govern their collective motion. Beyond species-specific details, our analysis suggests that force-mapping can identify the presence of alignment

interactions irrespective of if individuals are in a bounded domain (albeit using data from a very simple model here) or an unbounded domain. For the simple alignment-only model examined here, confinement within a bounded region resulted in a much clearer alignment response in our force maps than in an unbounded case, but this was coupled with overestimation or distortion of the region over which alignment occurred. A large portion of studies that have used force mapping have been of fish, or other aquatic species, confined to swim in laboratory aquaria (see for example [2, 8–11, 17, 37]. It might be reasonable to expect that for such aquaria studies that alignment interactions sought by force maps should clearly show such interactions if they are present, but that the force maps may overestimate the spatial range of alignment interactions. For practical reasons, there have been far fewer studies that have applied force mapping to data from species in effectively unbounded domains. One notable exception has been in the study of homing pigeons, as in [38]. Among a raft of force map analyses, [38] examined how individual pigeons adjusted their direction of motion as a function of the directions to a neighbouring pigeon and the angular difference in directions of motion (as per [10]), and more simply how individual pigeons adjusted their direction of motion as a function of the angular difference in direction of motion with their neighbour. The latter, simpler, measure suggested an underlying tendency to align with neighbouring pigeons, consistent with previous analysis of this species in flight via correlations in direction of motion in [39]. Applying the force maps that we have examined here to pigeon data may add more information about the approximate spatial extent over which this species aligns with neighbours. In broader terms, the additions to force mapping that we have examined here can help build a more complete picture of the interactions that drive collective animal motion, along with more nuanced understanding of the benefits and drawbacks of force mapping techniques.

## Supporting information

**S1 Data. MATLAB source codes for simulation models and force map analysis, as described in the main text and supplementary information.**
(ZIP)

**S1 File. An examination of force maps targeted at orientation interactions in moving groups.** Supplementary Information including details on force mapping, simulation models, supplementary tables, and supplementary. Figs and results referenced in the main text.
(PDF)

## Acknowledgments

We thank Norman Gaywood for his support for this project through his management of the Turing computational system at the University of New England, which was vital for the completion of this work. We also thank Geoff Bratt for his elegant method for constraining turning angles in our implementation of the zonal model, and for thoughtful feedback on this work by Ryan Lukeman, Mary Myerscough and Andrea Perna.

## Author Contributions

**Conceptualization:** Rajnesh K. Mudaliar, Timothy M. Schaerf.

**Formal analysis:** Rajnesh K. Mudaliar, Timothy M. Schaerf.

**Investigation:** Rajnesh K. Mudaliar, Timothy M. Schaerf.

**Methodology:** Rajnesh K. Mudaliar, Timothy M. Schaerf.

**Project administration:** Timothy M. Schaerf.

**Supervision:** Timothy M. Schaerf.

**Writing – original draft:** Rajnesh K. Mudaliar.

**Writing – review & editing:** Rajnesh K. Mudaliar, Timothy M. Schaerf.

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
