## [Decision Letter · Decision Letter 0]

3 Feb 2023

PONE-D-22-34379An examination of force maps targeted at orientation interactions in moving groupsPLOS ONE

Dear Dr. Mudaliar,

Thank you for submitting your manuscript to PLOS ONE. After careful consideration, we feel that it has merit but does not fully meet PLOS ONE’s publication criteria as it currently stands. Therefore, we invite you to submit a revised version of the manuscript that addresses the points raised during the review process.

 You will find in the attached report a list of recommendations that you are encouraged to follow in order to improve your paper.

We look forward to receiving your revised manuscript.

Kind regards,

Thierry Goudon

Academic Editor

PLOS ONE

Journal Requirements:

"This work was supported in part by funding from the Australian Research Council's

Discovery Project scheme, under project code DP190100660."

"This work was supported in part by funding from the Australian Research Council's

Discovery Project scheme, under project code DP190100660."

"We thank Norman Gaywood for his support for this project through his management of the Turing computational system at the University of New England, which was vital for the completion of this work. We also thank Geoff Bratt for his elegant method for constraining turning angles in our implementation of the zonal model, and for thoughtful feedback on this work by Ryan Lukeman, Mary Myerscough and Andrea Perna. This work was supported by funding from the Australian Research Council under project DP190100660."

"This work was supported in part by funding from the Australian Research Council's Discovery Project scheme, under project code DP190100660."

Reviewers' comments:

Reviewer's Responses to Questions

**Comments to the Author**

1. Is the manuscript technically sound, and do the data support the conclusions?

Reviewer #1: Yes

2. Has the statistical analysis been performed appropriately and rigorously? 

Reviewer #1: Yes

3. Have the authors made all data underlying the findings in their manuscript fully available?

Reviewer #1: Yes

4. Is the manuscript presented in an intelligible fashion and written in standard English?

Reviewer #1: Yes

5. Review Comments to the Author

Reviewer #1: My detailed review has been attached as a PDF file. This paper is potentially suitable to be published in PLOS One once my questions/issues included in my review are properly addressed. I would like to highlight that the authors have provided full details of their research on the SI, as well as the code, something I consider a good scientific practice.

6. PLOS authors have the option to publish the peer review history of their article (what does this mean?). If published, this will include your full peer review and any attached files.

Reviewer #1: No

---

## [Author Response · Author response to Decision Letter 0]

19 Apr 2023

Dear Professor Goudon,

Thank you for handling our manuscript, and thank you to our reviewer for providing helpful feedback which we have used to revise our manuscript. 

The most substantial technical change we have made to the manuscript is in response to the reviewer’s third question (reproduced below, with our detailed responses). The reviewer noted that we had not used consistent group sizes across simulation models for the work presented in our previous submission. We had presented results for N = 10 individual simulations for the alignment-only and ODE models, and for N = 25 individuals for the zonal model. To make our presentation more consistent in this respect, we re-ran our analysis of zonal model data for groups of N = 10, with these results now presented in the manuscript, as opposed to those for groups of N = 25. This affects the detail of a large number of the figures that illustrated results from analysis of zonal model data throughout the paper, but our broad results remain unchanged. 

In terms of the overall presentation of the paper, our largest additions are in response to the reviewer’s first, second and sixth points, which suggested summarising some of our results via a table (we have now added two tables relating to force maps in the discussion section), more explicitly connecting our work to real world species and suggestions for improving force mapping (we have added about two pages of text to the discussion to try to address these latter two points). Our detailed responses to the reviewer follow in bold; line references in these responses are connected to the highlighted revised versions of the manuscript and SI.

Yours sincerely, 

Rajnesh Mudaliar.

Q1) I think the paper lacks of a clear, concise and succinct way of presenting the main results of the analysis presented in this work. What I mean is that a reader might not easily find the information on when the FM method provides a reliable way to infer properties of the social interactions of moving individuals. I know that this information is already in the paper, but it is rather scattered/distributed in the discussion, results and SI sections. Perhaps, the authors could provide a sort of list/table in the discussion section for this (not 100% sure this is the best option, perhaps the authors have a better idea). By doing so, I believe that the impact of this work might be increased.

Response: Thank you, this is a very good point coupled with a helpful suggestion. For this revision we have tried to collate the broad elements of our findings relating to force maps of Δθ/Δt(d, φ) and Δθ/Δt(x, y, φ) in two new tables (Tables 5 and 6, on pages 36 and 37 of the pdf version of our submission). Within the tables we outline features well or reasonably resolved via force mapping seeking alignment behaviour for each large data set, along with clear issues and inaccuracies. These tables are referenced within the discussion on lines 634-636 and 640. 

Q2) A second important point would be to relate the results provided here to real-life biological and/or synthetic systems. If a reader of this paper is working with, let's say, fish in a tank [bounded system], and another reader is working with moving groups of ungulates in the wild [unbounded system], what should they expect from the FM method based on the results presented in this paper by the authors? This might be information that could be added/merged to the table of Q1, and I would also suggest to add this in the discussion section as well.

Response: We have added a substantial new final paragraph to the paper that discusses the implications of our work in the context of real-world studies that have applied force mapping previously, especially work on fish and other aquatic creatures in bounded domains (laboratory aquaria) and work on species in unbounded domains, with a focus on pigeons. (Discussion, lines 756-804.)

Q3) Why did the authors use two different group sizes for the results presented for the Vicsek model [they used N = 10 individuals] and the results for the zone model [they used N = 25 individuals]? Is there an important reason for doing so? If so, this reason should be discussed in the paper. If not, I would suggest to run new simulations and modify the plots to have only one system size across all the paper, which could be fixed, for example, to N = 25.

Response: This is a good point. In our previous submission, we choose group sizes to focus on for each model that provided some form of indicative results for what we observed for that model data in particular, without considering the need for consistency in group size

In the original submission results relating to both the Vicsek model and the ODE model were derived from simulations with N = 10 individuals. To maintain consistency in group size across the models, we have now redone all of the calculations relating to the zonal model for groups of size N = 10, with the corresponding results presented throughout the paper. Our broad results derived from zonal model data with N = 10 remain largely unchanged in comparison to those for N = 25, except that blind zones may be apparent for the smaller group sizes (now noted on lines 561-564, with reference to the updated Figures 7 R and 8 C), and that all variations of model details relating to the presence and extent of the blind zone could lead to either emergent cohesive or parallel motion (reflected in panels A to J of an updated and expanded Figure 9). The figures updated as a result of this revision to calculations are Figure 2 (panels L and M), Figures 7-9 and Figures S10-S20 (Supplementary Information). Additional updated text relating to the size of groups and number of simulations performed appears on lines 308-312 of the Supplementary Information document.

Q4) The size of the plots and labels of most of the figures of the paper are very small.

I would suggest to increase the font size and the size of all the plots.

Response: Where possible we have tried to increase the font size for figures in the main text, especially Figures 2-4. We had some difficulties increasing the font size for other plots, as doing so to any extent meant that axis titles and labels from the multiple panels overlapped, and we had less control over this problem than we would have liked in MATLAB. We have provided high resolution tiff files of all figures with this revised submission of the paper.

Q5) In Figure 6, the authors show how the area of the region high alignment of the Vicsek model changes with increasing interaction radius for different system sizes. They report a simple linear regression, but visually to me it looks that the dependence might be rather non-linear, perhaps with some exponent larger than 1. Did the authors implement other non-linear regression methods? If so, please comment on why they did not included them on the paper. If not, I would suggest the authors to do it and interpret the resulting exponent. Also, in the same figure, there seems to be a slight dependency on the system size. Could the authors please comment on this and what would this mean when studying real-life large systems? [see Q1 and Q2].

Response: In our original submission, Figure 6 panels A and C illustrated the relationship between prescribed alignment interaction radius for the alignment-only model and measured areas of high alignment in our local alignment plots (these panels appear as panels A and B of a revised Figure 6 in the revised manuscript). We did not perform a linear regression for the data illustrated in these plots in our original submission. This was because we expected these plots to reflect something of the general relationship between the radius of a circle (r) and its area (πr^2), and indeed this was the case, with what appears to be an nonlinear (probably quadratic?) relationship, as observed by our reviewer. We have added some text to try to clarify this (on lines 498-503 and the caption for Figure 6, page 20). In addition, we have expanded Figure 6 so that it now does include all the data on which we applied linear regression, as detailed in Tables 1 to 4. We think that at least some elements of the system-size dependence that we observed in an unbounded domain (noted on lines 514-521 of the revised manuscript) could be specific to the much simplified alignment-only model. As a consequence, we have held back from inferring more general comments from this element of our results, in favour of our broader observations across models that suggest that the local alignment plots tend to reflect emergent patterns (especially when more complex/realistic sets of interactions drive collective motion); see lines 676-691.

Q6) Finally, do the authors have some concrete suggestions to improve the FM method?

For example, proposing a new way to aggregate data to produce more informative heat maps? For this, no exhaustive study is needed but rather the authors could use the knowledge they got from using the FM method to propose improvements as an outlook of the whole paper.

Response: We have some ideas relating to patchy data coverage, and mitigation of group patterns of movement affecting force maps that we have now included in the discussion (lines 707-746). As noted in our introduction, Heras et al. approached force mapping from a machine-learning perspective, and in the process developed a separate module within their machine-learning method for weighting interactions with different neighbours (very briefly mentioned on lines 95-98), which in effect deals with some of the issues of how data is aggregated in our much simpler approach to force mapping.

---

## [Editor Report · Decision Letter 1]

24 May 2023

An examination of force maps targeted at orientation interactions in moving groups

PONE-D-22-34379R1

Dear Dr. Mudaliar,

We’re pleased to inform you that your manuscript has been judged scientifically suitable for publication and will be formally accepted for publication once it meets all outstanding technical requirements.

Kind regards,

Thierry Goudon

Academic Editor

PLOS ONE
---

## [Editor Report · Acceptance letter]

31 May 2023

PONE-D-22-34379R1 

An examination of force maps targeted at orientation interactions in moving groups 

Dear Dr. Mudaliar:

I'm pleased to inform you that your manuscript has been deemed suitable for publication in PLOS ONE. Congratulations! Your manuscript is now with our production department. 

Kind regards, 

on behalf of

Dr. Thierry Goudon 

Academic Editor

PLOS ONE